# Unattached kinetochores drive their own capturing by sequestering a CLASP

Caroline Kolenda[1], Jennifer Ortiz[1], Marina Pelzl[1], Sarina Norell[1], Verena Schmeiser[1] & Johannes Lechner[1]

Kinetochores that are not attached to microtubules prevent chromosome missegregation via the spindle assembly checkpoint. We show that they also promote their own capturing. Similar to what governs the localization of spindle assembly checkpoint proteins, the phosphorylation of Spc105 by Mps1 allows unattached kinetochores to sequester Stu1 in cooperation with Slk19. The withdrawal of Stu1, a CLASP essential for spindle integrity, from microtubules and attached kinetochores disrupts the organization of the spindle and thus allows the enhanced formation of dynamic random microtubules that span the nucleus and are ideal to capture unattached kinetochores. The enhanced formation of nuclear random microtubules does not occur if Stu1 sequestering to unattached kinetochores fails and the spindle remains uncompromised. Consequently, these cells exhibit a severely decreased capturing efficiency. After the capturing event, Stu1 is relocated to the capturing microtubule and prevents precocious microtubule depolymerization as long as kinetochores are laterally or incompletely end-on attached.

[1] Biochemie-Zentrum der Universität Heidelberg, INF 328, 69120 Heidelberg, Germany. Correspondence and requests for materials should be addressed to J.L. (email: johannes.lechner@bzh.uni-heidelberg.de)

Reliable chromosome segregation depends on the correct attachment of kinetochores (KTs) to KT-microtubules (kMTs). In budding yeast, chromosomes detach during S-phase[1]. Subsequently, the unattached KTs (uaKTs) are captured by nuclear random MTs (nrMTs) in prometaphase[2,3]. First, KTs attach to the MT lattice and this frequently suffices to transport the chromosomes to the spindle pole body (SPB). If the MT depolymerizes to an extent that the plus end reaches the KT, it can establish end-on attachment and move with the depolymerizing MT plus end. Two KT components, the Ndc80 complex (Ndc80c) and the Dam1 complex, are essential to achieve and maintain KT end-on attachment.[2,4] In contrast to this, the Ndc80c alone is sufficient for lateral attachment[2]. The transition from lateral to end-on attachment during the capturing process includes the risk of KT dissociation. It appears thus beneficial to prevent the depolymerization of the MT beyond a point that makes this transition necessary and promote the re-polymerization of this MT. Stu2, a member of the XMAP215 family, is involved in this process[2,5].

The putative MT rescue factor Stu1, a member of the CLASP family, may also facilitate KT capturing[6]. CLASPs have TOGL domains that are required for their rescue function[7]. Stu1 has two N-terminal TOGL domains (TOGL1,2). Only TOGL2 provides direct rescue activity, whereas TOGL1 serves as a KT-binding domain[8] (Fig. 1a). In metaphase, Stu1 has at least two roles. First, it stabilizes interpolar MTs (ipMTs) via a direct interaction that involves the TOGL2 domain and a basic serine-rich unstructured region (ML). Second, it localizes to the KTs via TOGL1 and ML, and stabilizes kMTs[8]. Consequently, Stu1 is essential for the formation of a metaphase spindle[8,9]. In prometaphase, Stu1 is sequestered at uaKTs and this depends on the outer KT proteins Ndc80 and Spc105, the TOGL1 domain of Stu1, and a presumably unstructured region of Stu1, the C-terminal loop (CL)[6,8].

The mechanism that governs the sequestering is unclear. Furthermore, the benefit of this dramatic effect has stayed obscure. Here we show that the localization of Stu1 at uaKTs depends on the activity of the protein kinase Mps1 and on Slk19, a protein that supports spindle stability[10,11] and the clustering of uaKTs[12]. Importantly, we demonstrate that the sequestering of Stu1 causes a restructuring of the nuclear MT network that favors the capturing of uaKTs. Furthermore, Stu1 delivered via an uaKT to a capturing MT reduces the frequency of plus-end attachment distal to the SPB and prevents MT depolymerization if the transition from lateral to end-on attachment is compromised.

## Results

**Slk19 facilitates sequestering of Stu1 at uaKTs.** As Slk19 has been described to promote the clustering of uaKTs[12], we observed the cellular Slk19 localization in the presence of uaKTs and found that it resembles that of Stu1 (Fig. 1b,c). Slk19 strongly accumulated at uaKTs after nocodazole treatment and could hardly be detected at attached KTs and short MTs in the vicinity of the SPB in wild-type (WT) cells. In contrast, other spindle-localized proteins tested (Stu2, Bim1, Bik1, Kar3, Cin8, Fin1, Kip1, and Ase1) did not exhibit accumulation at uaKTs (Supplementary Fig. 1). In Stu1-depleted cells, Slk19 localized weakly to attached KTs and uaKTs indiscriminately (Fig. 1d). Although a basal level of Slk19 can localize at uaKTs in the absence of Stu1, Slk19 accumulation depends on Stu1. Conversely, in Δslk19 cells, Stu1 localized to both uaKTs and attached KTs (plus MTs) at the SPB indiscriminately (Fig. 1e). Thus, although Stu1 sequestering requires Slk19, a basal amount of Stu1 can localize to uaKTs in the absence of Slk19.

Stu1 forms a homodimer[8] and Slk19 a homotetramer[13] in solution. A model how Slk19 might facilitate the sequestering of Stu1 and vice versa via a three-dimensional array of alternating Stu1 and Slk19 molecules is shown in Fig. 1n. In agreement with this, Stu1 requires dimerization for efficient sequestering[8]. Furthermore, it is in agreement with the fact that not only Slk19[12] but also Stu1 supports the clustering of uaKTs (compare Fig. 1b and d, and Supplementary Fig 2). Notably, we were not able to co-immunoprecipitate Slk19 with Spc105 or Stu1 when cells were treated with nocodazole. This may indicate that the oligomerized complex is not stable under the experimental conditions used for cell lysis and immunoprecipitation, or that Stu1 and Slk19 interact indirectly within this complex.

**The CL of Stu1 facilitates oligomerization of Stu1 and Slk19.** If the CL domain is deleted, Stu1 can still localize to uaKTs but sequestering is defective[8] (Fig. 1f). Thus, the phenotype of stu1ΔCL cells reflects that of Δslk19 cells. Slk19 accumulation at uaKTs is clearly defective in stu1ΔCL cells (Fig. 1g). Furthermore, based on the above model (Fig. 1n), Stu1ΔTOGL1ΔD4 (that cannot bind to uaKTs directly, support oligomerization or form heterodimers with WT Stu1[8]) should bind to the Slk19 endpoints of an array formed in a STU1 WT background. This is indeed the case (Fig. 1h). However, Stu1ΔTOGL1ΔCLΔD4 fails to localize to uaKTs even in a STU1 WT background (Fig. 1i). Taken together, this suggests that the CL domain of Stu1 facilitates an oligomerization of Stu1 with Slk19, which drives sequestering.

**Sequestering of Stu1 at uaKTs depends on Mps1 activity.** How does the absence of KT-MT interaction translate into Stu1 sequestering? Likewise, SAC proteins localize specifically to uaKTs[14]. Here, the absence of KT–MT interaction allows Mps1 to phosphorylate the KT protein Spc105, thus initiating the assembly of SAC proteins[15–17]. In contrast to WT cells, mps1-as1 cells[18] treated with the inhibitor 1NM-PP1 showed no detectable Stu1 at uaKTs (Fig. 1j). On the other hand, the basal localization of Slk19 was not dependent on Mps1 (Fig. 1k). We therefore suggest that Mps1 activity promotes the direct interaction of Stu1, but not Slk19, with the KT, and that Stu1 at uaKTs initiates Stu1–Slk19 oligomerization (see discussion).

Stu1 sequestering at uaKTs depends on the KT protein Spc105[8] (Supplementary Fig. 3a). The phosphorylation of Spc105 by Mps1 at six MELT consensus sites initiates SAC signaling[19]. Mutating these sites to alanine (spc105-6A) strongly interfered with Stu1 sequestering (Fig. 1l), indicating that their phosphorylation is essential for Stu1 sequestering. Consistent with the fact that Stu1 localizes to uaKTs and (in the absence of uaKTs) to attached KTs[8], we found that Stu1 co-purifies with Spc105 irrespective of whether cells were treated with nocodazole or not (Supplementary Fig. 4). Furthermore, this may indicate that Stu1 localizes in the vicinity of Spc105 in both cases. Stu1 co-purified with similar (moderate) amounts with Spc105 in the presence and absence of nocodazole. This probably indicates that after nocodazole treatment we were not able to purify oligomerized Stu1 but rather the basal amount of Stu1 found at uaKTs. As mentioned above, the Stu1/Slk19 oligomer may not be stable under the cell lysis and immunoprecipitation conditions applied. Stu1 also co-purified with Spc105-6A when cells were treated with nocodazole (Supplementary Fig. 4). As Stu1 localizes exclusively to attached KTs in nocodazole-treated spc105-6A cells (Fig. 1l), this indicates that Stu1 binding to attached KTs does not depend on Spc105 phosphorylation at the six MELT consensus sites. This is consistent with the fact that these sites are not phosphorylated when KTs are attached to MTs[19].

**Stu1 is sequestered at attached KTs upon ectopic Mps1 binding**. The lack of KT–MT interaction is thought to cause a change within the KT structure that allows Ndc80c-bound Mps1 to phosphorylate Spc105[17]. Does this also regulate Stu1 sequestering at uaKTs? As described[17], we localized Mps1, rapamycin-controlled, to the C terminus of Spc105. This caused the sequestering of Stu1ΔML at the attached KTs of metaphase-arrested cells (Fig. 1m). Notably, Stu1ΔML can be sequestered at uaKTs but cannot bind to attached KTs or MTs in control cells[8] (Fig. 1m). Thus, the Ndc80c-independent localization of Mps1 in the vicinity of Spc105 induces Stu1 sequestering despite MT–KT interaction. We therefore propose that a change in the

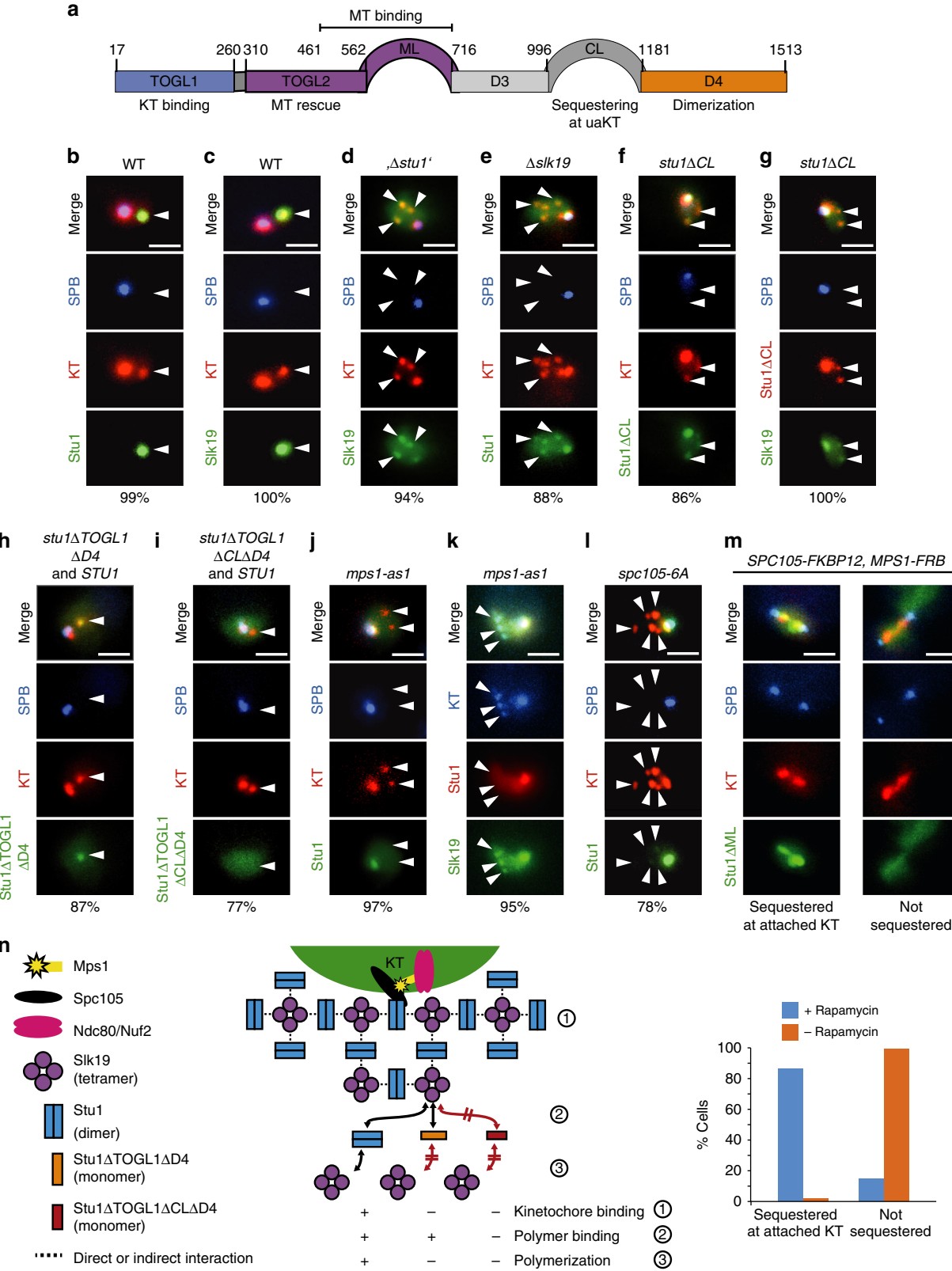

accessibility of Spc105 for Mps1 phosphorylation translates KT–MT detachment not only in SAC activation but also in Stu1 sequestering.

Taken together, the principles that govern the localization of Stu1 to uaKTs resemble that of SAC activation. Notably, however, the sequestering of Stu1 does not depend on the SAC proteins Bub1 and Mad2[6] or Bub3, Mad1, and Mad3 (Supplementary Fig. 3b).

**Stu1 sequestering at uaKTs depletes it from the spindle.** After nocodazole treatment, it is impossible to observe the consequence of Stu1 sequestering on spindle MTs. We therefore evoked uaKTs by inactivation and subsequent reactivation of CEN5 in the presence of an intact spindle[2]. As observed for nocodazole treatment, uaKTs produced by KT reactivation sequestered Stu1. Starting with Stu1 localizing exclusively at the spindle region, Stu1 accumulated at the uaKT with time after KT reactivation (Fig. 2a). Twenty percent of the uaKTs that were not captured within 20 min after reactivation did not acquire Stu1, possibly because their KTs failed to reassemble. In the remaining cells, the uaKT accumulated within 20 min on average 1.3 times the amount of Stu1 that had localized to the spindle region at the beginning of the experiment. Within this time period the amount of Stu1 localized to the spindle region dropped on average to 40% of its original level. Notably, after 20 min the additive amount of Stu1 at the uaKT and the spindle was higher than the amount of spindle-localized Stu1 at the beginning. Furthermore, 7 min after reactivation, the uaKT had sequestered on average a substantial amount of Stu1, although the amount of Stu1 at the spindle was unchanged. This indicates that Stu1 sequestering at uaKTs withdraws Stu1 also from a soluble pool. The amount of Stu1 localizing at the spindle remained unchanged (Fig. 2b, c, d) in experiments performed with cells unable to sequester Stu1 (stu1ΔCL, Δslk19, or Mps1-depleted cells). Taken together, these results strongly suggest that the sequestering of Stu1 at uaKTs causes the withdrawal of Stu1 from spindle MTs and attached KTs probably via a soluble pool that is in equilibrium with the spindle-localized Stu1. Earlier fluorescence recovery after photobleaching analysis of spindle and uaKT-localized Stu1 supports this[6].

**Sequestering of Stu1 causes the enhanced formation of nrMTs.** The sequestering of Stu1 correlated with the collapse of the spindle. The capturing of the uaKT and the subsequent re-localization of Stu1 to the spindle region correlated with the recovery of a stable metaphase spindle (Fig. 3a). Surprisingly, sequestering of Stu1 also correlated with the appearance of an increased number and length of MTs that were not spindle MTs (Fig. 3a). Importantly, these MTs included to a large extent nrMTs as shown by Dad1 localization (Fig. 3b, c). To quantify this observation, we compared the average nrMT length per timepoint. For most cells with an uaKT that had sequestered Stu1, this was considerably higher than for cells without an uaKT (Fig. 3d). Furthermore, when averaged over all cells, due to the increased occurrence and length of nrMTs (Fig. 3f, g), the MT length per timepoint was 20-fold higher in cells with an uaKT than in cells without one (Fig. 3e).

As shown above, stu1ΔCL and Mps1-depleted cells failed to sequester Stu1 and left the spindle largely uncompromised. The nrMT length per timepoint in cells with an uaKT was severely lower in stu1ΔCL and Mps1-depleted than in WT cells (Fig. 4). For stu1ΔCL cells, it reflected the data of WT cells without an uaKT (Fig. 3d–g). For Mps1-depleted cells, it was somewhat higher but reflected the data of Mps1-depleted cells without an unattached KT5 (Fig. 4b, c; see below for further discussion). Taken together, these experiments show that the sequestering of Stu1 at uaKTs causes the enhanced formation of nrMTs.

**Depriving the spindle of Stu1/Slk19 promotes nrMTs formation.** Is the removal of Stu1 or Slk19 from the spindle sufficient to enhance the formation of nrMTs? The formation of nrMTs was strongly enhanced in metaphase-arrested Δslk19 or Stu1-depleted cells, even in the absence of uaKTs. The average nrMT number and length was dramatically increased in comparison with that in WT cells without or even with uaKTs (Fig. 5a–d). This is consistent with the idea that the efficiency at which nrMTs form is inversely correlated to the amount of Stu1 or Slk19 available to stabilize the spindle. Furthermore, it shows that removal of Stu1 or Slk19 from the spindle is sufficient to drive the formation of nrMTs. Importantly, the dynamic instability (growth rate, shrinkage rate, rescue, and catastrophe frequency) of nrMTs produced as a result of Stu1 depletion or Stu1 sequestering was similarly high (Fig. 5e, f). Thus, Stu1 apparently has no major role in regulating the stability of nrMTs before they engage in KT capturing.

**Defective Stu1 sequestering compromises capturing.** The enhanced appearance of dynamic MTs spanning the nucleus as a result of uaKTs might enhance the capturing of uaKTs. stu1ΔCL cells (that are unable to sequester Stu1 and have no enhanced formation of nrMTs) were severely compromised in the capturing of uaKTs (Fig. 6a) in comparison with WT cells. stu1ΔCL cells carrying an additional STU1 WT allele were similarly compromised, indicating that Stu1 sequestering per se is insufficient to support capturing if the spindle remains intact (via Stu1ΔCL). In Mps1-depleted cells, uaKTs were captured also clearly less efficiently than in WT cells but slightly better than in stu1ΔCL cells. This observation correlates well with the fact that

**Fig. 1** Dependencies of Stu1 and Slk19 localization and sequestering to uaKTs. **a** Domain structure and function of Stu1[8]. **b–m** Genotypes of the strains revealing the fluorescent fusion proteins are listed in Supplementary Table 1. The percentage of cells revealing the depicted phenotype is indicated. Statistics see Supplementary Table 3. Bars, 2 μm. **b–l** Cells with the indicated genetic background were analyzed 3–4 h after the release from G1 into nocodazole. White arrowheads indicate uaKTs. **b** Stu1 is sequestered at uaKTs[6]. **c** Slk19 accumulates at uaKTs. **d** The accumulation of Slk19 at uaKTs depends on Stu1., Δstu1' indicates that Stu1 was depleted. **e** The sequestering of Stu1 at uaKTs depends on Slk19. **f** The CL domain of Stu1 is required for Stu1 sequestering[8]. **g** The CL domain of Stu1 is required for the accumulation of Slk19. **h** Stu1ΔTOGL1D4 can localize to uaKTs in the presence of WT Stu1. **i** Stu1ΔTOGL1CLΔD4 cannot localize to uaKTs even in the presence of WT Stu1. **j** The localization of Stu1 depends on Mps1. 1NM-PP1 was added after the G1 release. **k** Sequestering of Slk19 but not the basal localization of Slk19 depends on Mps1. Cells were treated as in **j**. **l** Mutating the six MELT Mps1 phosphorylation sites in Spc105[19] to alanine (spc105-6A) prevents Stu1 sequestering. WT Spc105 in the background was depleted during the G1 arrest and Cdc20 was depleted with the G1 release to prevent the SAC-deficient cells from progressing into anaphase. **m** Stu1ΔML is sequestered at attached KTs when Mps1 is localized to Spc105 ectopically. Cells were arrested in metaphase by Cdc20 depletion and rapamycin was added as indicated. WT Stu1 was present in the background. **n** Model depicting the sequestering of Stu1 at uaKTs. In the absence of KT-MT interaction, Ndc80c-localized Mps1 phosphorylates Spc105 and thus prompts Stu1 localization to uaKTs. A putative conformational change that involves the CL domain triggers the interaction of Stu1 and Slk19. Propagated conformational changes in Slk19 and Stu1 lead to co-polymerization. Stu1ΔTOGL1D4, which is unable to bind to uaKTs alone[8], can bind to the Slk19 endpoints of the polymer, whereas Stu1ΔTOGL1D4ΔCL cannot (see **h** and **i**)

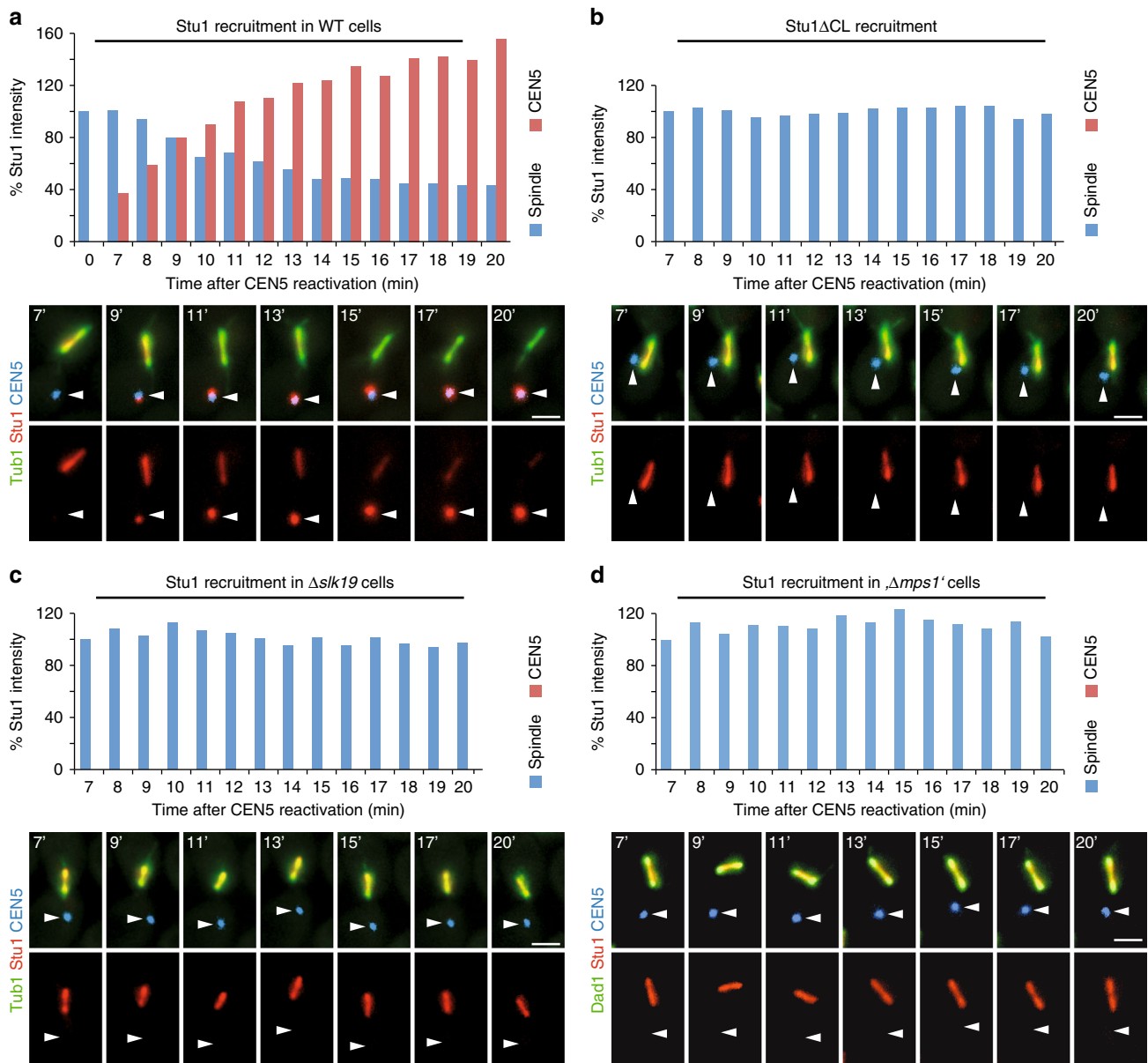

**Fig. 2** Sequestering at uaKTs withdraws Stu1 from spindle MTs and attached KTs. **a**–**d** Genotypes of the used strains are listed in Supplementary Table 1. Cells with the indicated genotypes were arrested in metaphase. Then the centromere DNA of chromosome V was inactivated and subsequently reactivated, thus producing an uaKT of chromosome V (white arrowheads). The upper panel of micrographs represents the overlay of the three signals, the lower panel the Stu1 signal. The average amount of Stu1 that localized to uaKTs or the spindle region (spindle) at the indicated timepoints after KT reactivation is shown for one of two independent experiments. Number of analyzed cells are shown in Supplementary Table 3. The amount of Stu1 measured for the spindle region 7 min after KT reactivation was set to 100%. Bars, 2 μm. **a** Stu1 is sequestered at uaKTs and is removed from the spindle and attached KTs. **b** Stu1ΔCL fails to get sequestered at uaKTs and remains at the spindle and attached KTs. WT Stu1 in the background was depleted. **c** In the absence of Slk19, Stu1 sequestering fails and Stu1 remains at the spindle and attached KTs. **d** In the absence of Mps1, Stu1 sequestering fails and Stu1 remains at the spindle and attached KTs. Mps1 was depleted during the metaphase arrest ('ΔMps1')

these cells form more nrMTs than *stu1ΔCL* cells but less than WT cells (see above). In contrast, Δ*slk19* or Stu1-depleted cells, which exhibit a permanently enhanced formation of nrMTs in metaphase, captured uaKTs at least as efficient as WT cells. We therefore suggest that the increased formation of nrMTs promotes the capturing of uaKTs, and that uaKTs advance their capturing by enhancing the formation of nrMTs via sequestering of Stu1 and Slk19 (Fig. 6b). We suggest that this strategy operates particularly in prometaphase to keep the SPBs in close proximity and thus provide nrMTs till all uaKTs are captured. This may be especially important if KTs remain unattached for a prolonged time. In agreement with this, we found that at a given timepoint

in prometaphase about 26% of the cells exhibited uaKTs with a substantial amount of sequestered Stu1 (Supplementary Fig. 5). Notably, this does not exclude that the remaining 74% of cells might have shown uaKTs with sequestered Stu1 at other timepoints in prometaphase.

**Post capturing Stu1 and Slk19 go to MTs and move to the SPB**. The interaction with the MT lattice is the first step in KT capturing. Subsequently, KTs reach the SPB predominantly via lateral gliding and, to a smaller extent, at the plus end of a depolymerizing MT[2,3,20]. During the interaction with the MT

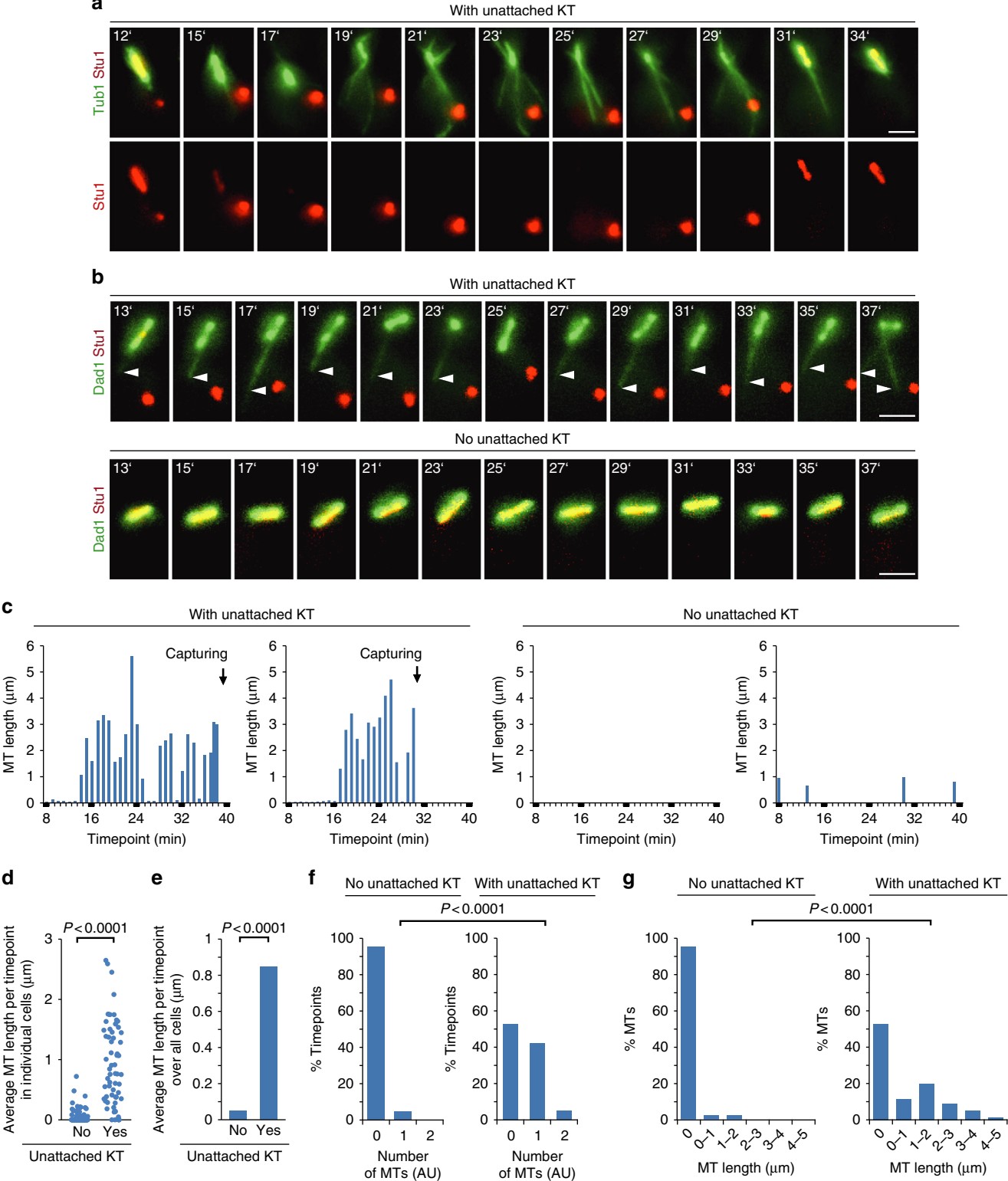

**Fig. 3** Sequestering of Stu1 at uaKTs enhances the formation of nrMTs. **a–g** Genotypes of the used strains are listed in Supplementary Table 1. **a** Sequestering of Stu1 at uaKTs results in spindle collapse and drives the formation of non-spindle MTs. Capturing of the uaKT allows the relocalization of Stu1 to the spindle region and spindle stabilization. Cells were analyzed at the indicated timepoints after KT reactivation as described in Fig. 2. MTs were visualized via GFP-Tub1. Bar, 2 μm. **b** Sequestering of Stu1 at uaKTs drives the formation of nrMTs. Experiment as in **a**, but revealing nuclear MTs via Dad1-GFP localization. Arrowheads indicate nrMTs. **c** Occurrence and length of nrMTs for four representative cells observed as in **b**. **d–g** Cells were observed as in **b** and the number and length of nrMTs were determined. Cells were analyzed from the timepoint when the uaKT had sequestered 55–75% of the total localized Stu1 to the timepoint before the uaKT interacted with a capturing MT or if the uaKT was not captured till the end of the film. Cells without an uaKT were analyzed over 15 timepoints. Statistics, see Supplementary Table 3

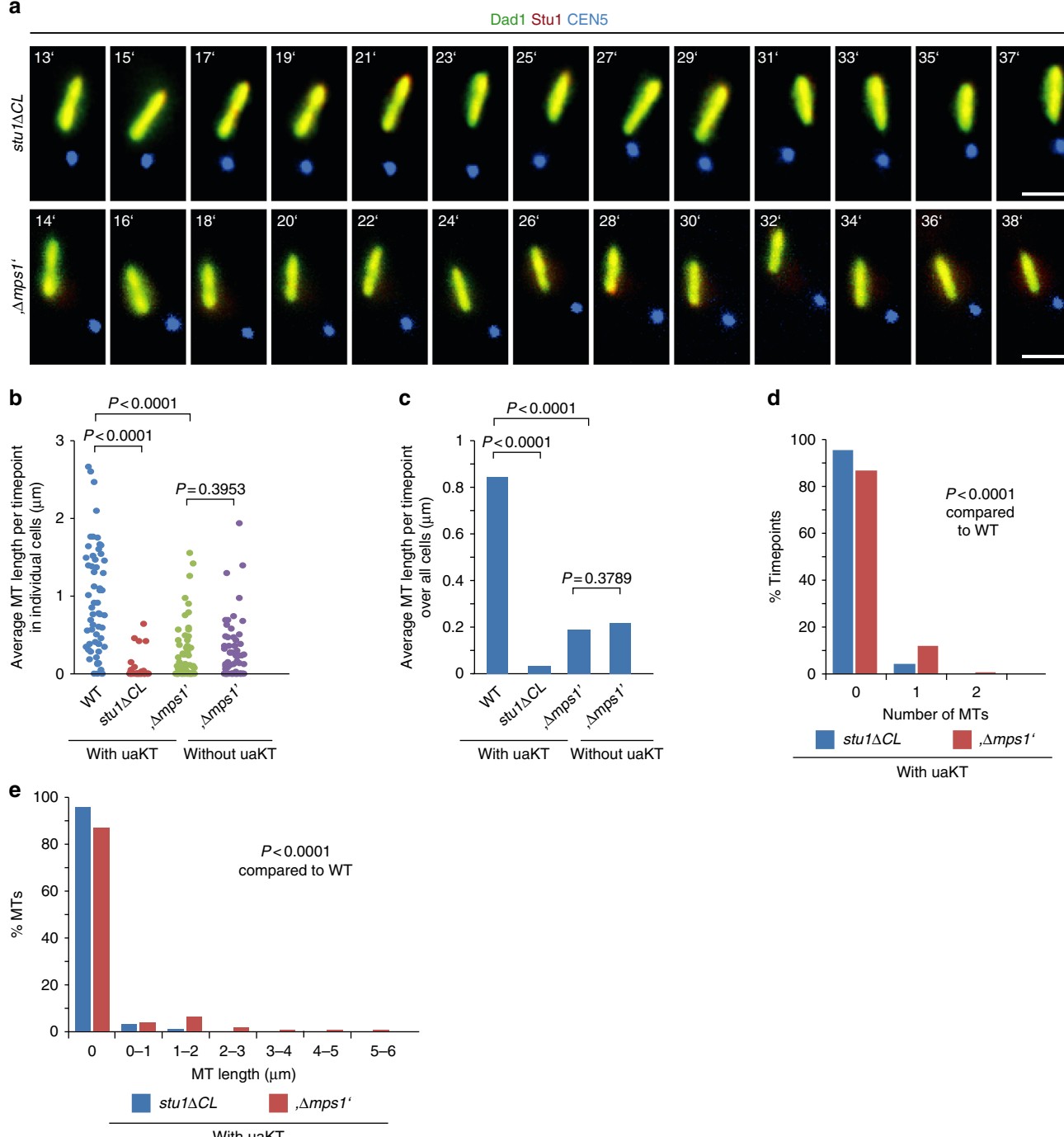

**Fig. 4** The enhanced formation of nrMTs fails if the sequestering of Stu1 at an uaKT is defective and the spindle remains uncompromised. **a–e** The experiment with *stu1ΔCL* and Mps1-depleted cells (genotypes see Supplementary Table 1) was performed as described in Fig. 2 and visualized as in Fig. 3b. **b–e** The number and length of nrMTs were determined. Cells with an uaKT were observed for 15 timepoints or until the uaKT was captured. The data corresponding to WT cells with an uaKT, as shown in Fig. 3d and e, has been included for comparison. Statistics, see Supplementary Table 3

lattice at least part of Stu1 and Slk19 frequently dissociated from the KT and colocalized on the capturing MT as clusters that moved towards the SPB (Fig. 7a,b). In 58% of the transport events, one of these clusters localized between the KT and the MT plus end till the KT was close to the SPB (Fig. 7c). In 30% of the events, all detectable Stu1 colocalized with the KT representing either KT- or MT-associated Stu1. In the vicinity of the SPB, Stu1 usually overtook and/or dissociated mostly from the KT, allowing Stu1 to stabilize the spindle before the KT achieved bipolar attachment.

**Stu1 minimizes the need for SPB-distal end-on conversion.** Stu1 most likely represents a MT rescue factor[7,8]. Thus, the strategic positioning of uaKT-derived Stu1 on a capturing MT might prevent the depolymerization of the MT so that KTs won't face the MT plus-end and risk detachment. In WT cells, KTs rarely established end-on attachment distal to the SPB (Fig. 8a), because the MT was rescued in advance (Fig. 8d, g). However, MT rescue was less frequent in Stu1-depleted and *Δslk19* cells (Fig. 8d, g). In agreement with this, captured KTs were found more frequently near MT plus ends distant to the SPB (Fig. 8i)

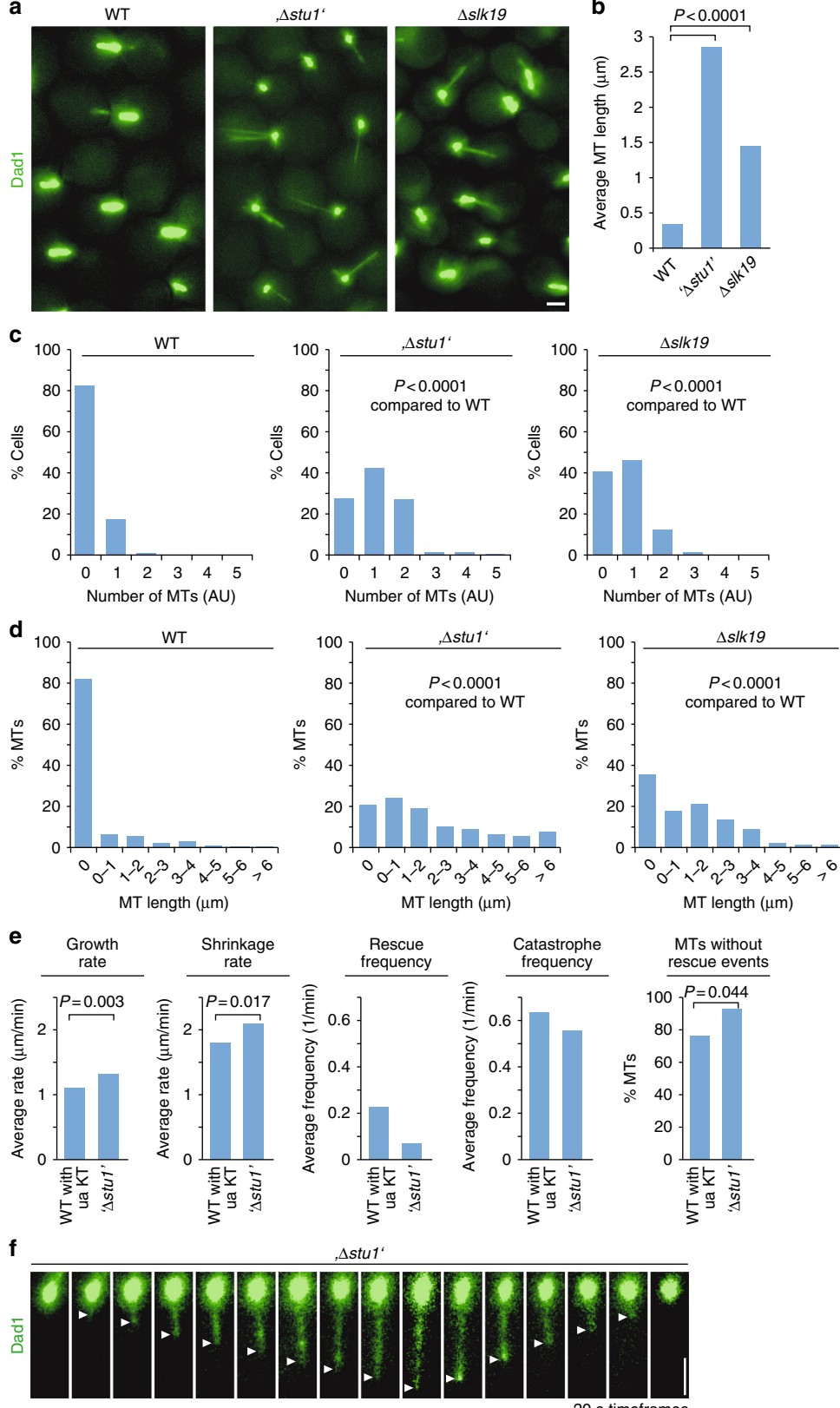

**Fig. 5** Compromising the metaphase spindle stability is sufficient to enhance the formation of dynamic nrMTs. **a–d** Cells with the indicated genotype (see Supplementary Table 1) were arrested in metaphase, nrMTs were visualized via Dad1-GFP and their length and number was determined. **e**, **f** nrMTs that are formed as a result of Stu1 sequestering at uaKTs or Stu1 depletion exhibit in both cases high dynamic instability. Cells were arrested in metaphase and nrMTs were visualized via Dad1-GFP. WT cells with an uaKT (produced as in Fig. 2) that had sequestered at least 55% of the localized Stu1, or Stu1-depleted ('Δstu1') cells were analyzed at 20 s intervals. **b–e** Statistics, see Supplementary Table 3. **f** White arrowheads mark the MT plus ends

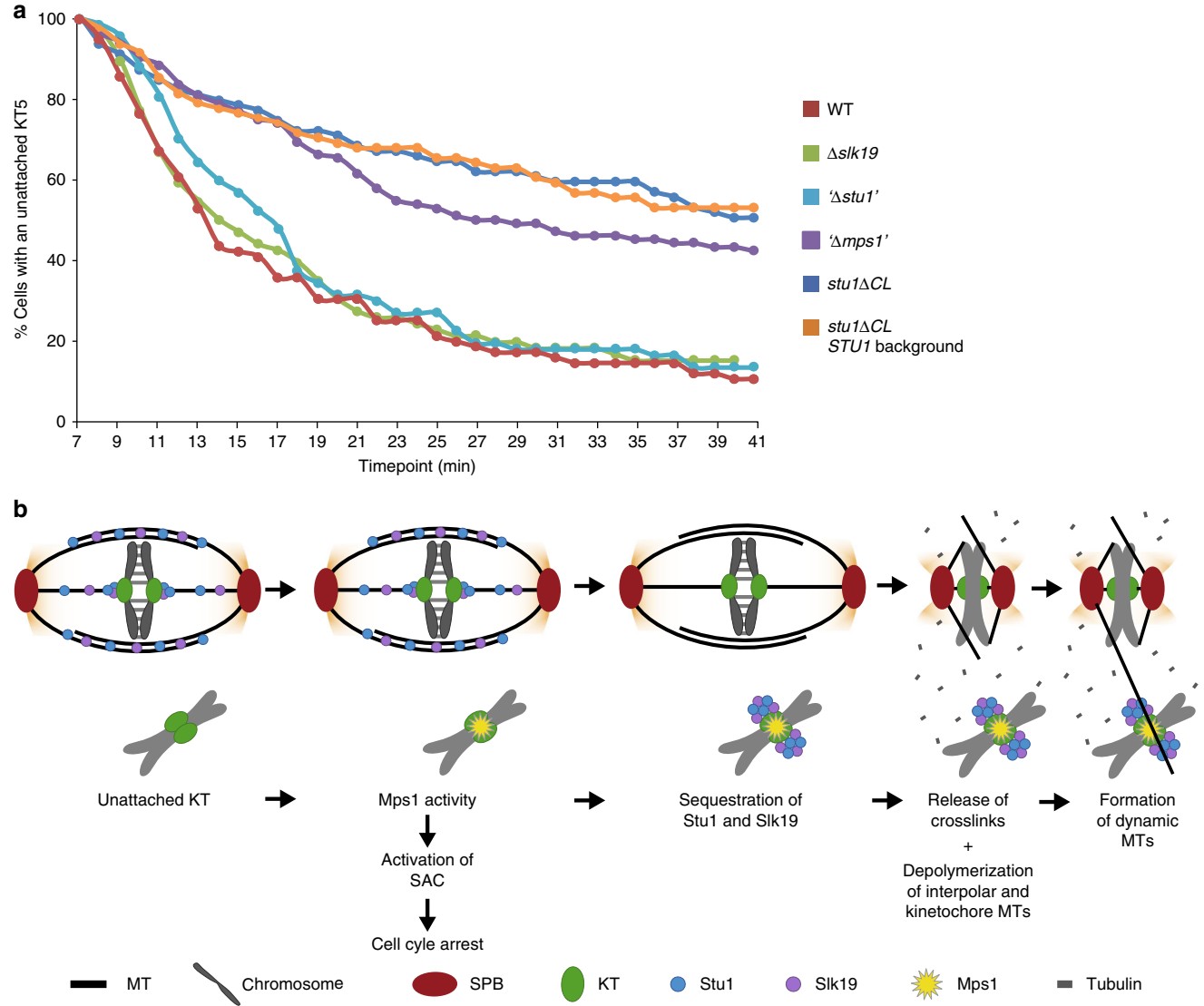

**Fig. 6** The enhanced formation of nrMTs facilitates the capturing of uaKTs. **a** Compromising the mitotic spindle by Stu1 removal is important for efficient capturing of uaKTs. Cells with the indicated genotypes (see Supplementary Table 1) were arrested in metaphase and uaKTs were produced as described in Fig. 2. Cells were observed in 1 min time intervals and the number of cells with an uaKT was determined. The number of cells with an uaKT observed 7 min after KT reactivation (see Supplementary Table 3) was set to 100%. The data shown is the result of one out of two independent experiments performed. Both had a very similar outcome. **b** Proposed model how sequestering of Stu1 and Slk19 enhances the formation of nrMTs, and thus promotes capturing of uaKTs. Mps1 localized at an uaKT induces the binding of Stu1 to this uaKT. This initiates the sequestering of Stu1 and Slk19 at the uaKT. Consequently, Stu1 and Slk19 are withdrawn from the spindle resulting in the depolymerization of kMTs and ipMTs. The free tubulin produced in this way then promotes the formation of dynamic nrMTs that scan the nucleus and facilitate the capturing of the uaKT

and experienced more plus-end transport or detachment (Fig. 8a). Furthermore, the plus-end attachment was established on average at a greater distance from the SPB than in WT cells (Fig. 8j). We thus suggest that depositing uaKT-derived Stu1 onto the capturing MT after lateral attachment protects KTs from premature plus-end encounter. As Stu1 and KTs are frequently transported in sync, this protection persists till Stu1 overtakes the KT close to the pole. This minimizes the need of SPB-distal end-on conversion that may involve the risk of detachment[5] (Fig. 8m).

**Stu1 halts MT depolymerization if KT end-on attachment fails**. Does Stu1 have a role during the conversion from lateral to end-on attachment of a captured KT? In *dam1ΔC* cells, this step is compromised[4] and the KT frequently remains in the MT plus-end region (stand-still), whereas the MT remains stable or even

polymerizes (Fig. 8l). However, in *dam1ΔC* cells depleted of Stu1 or *dam1ΔC Δslk19* cells, the stand-still phenotype was minimal. Instead, an increased number of transport and detachment events (Fig. 8k) occurred. This indicates that Stu1 residing in the plus-end region in *dam1ΔC* cells (Fig. 8l) prevents MT depolymerization and KT detachment when the end-on attachment is compromised.

## Discussion

Our work describes a novel mechanism that enables uaKTs to promote their own capturing by sequestering the CLASP Stu1. The principles that initiate Stu1 sequestering appear similar to those that activate the SAC[14,17,19,21,22]: Localized to KTs via Ndc80c, Mps1 phosphorylates Spc105 at MELT sites in the absence of KT-MT interaction. This allows the localization of

Stu1 to uaKTs. Notably, Stu1 localization to attached KTs does not depend on Spc105 phosphorylation at the six MELT sites but depends on MT interaction[8]. The localization of Stu1 to uaKTs does not depend on the SAC proteins Bub1/Bub3 that directly interact with phosphorylated Spc105[19]. Further details, how Stu1 interacts with uaKTs have to be clarified by future (in vitro) analysis. Various possibilities can be envisioned. Stu1 may bind directly to phosphorylated MELT sites in Spc105 as described for

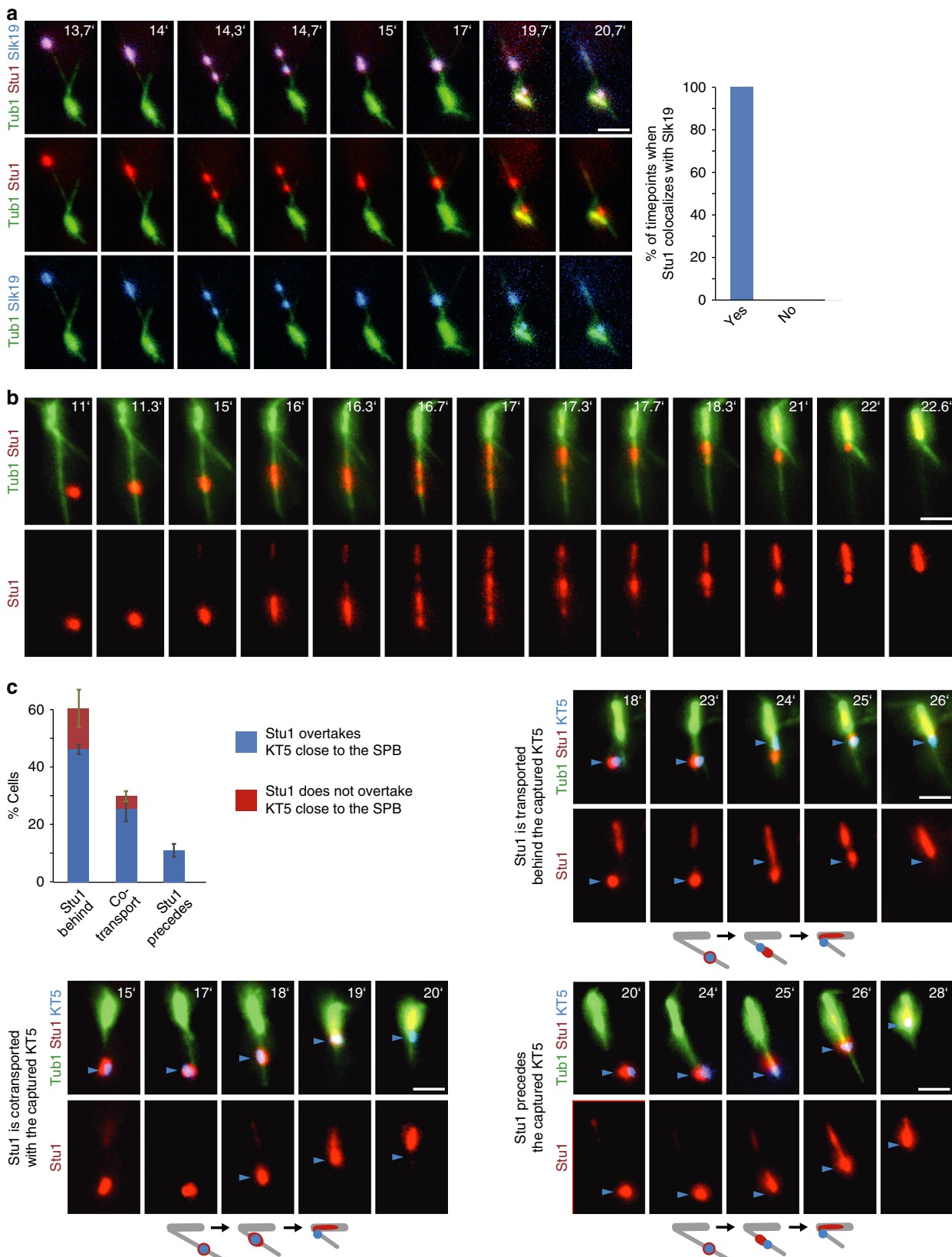

hBub1/hBub3[23]. Alternatively, Stu1 may bind to a conformation of Spc105 or a neighboring KT protein induced by the phosphorylation event. In any case, SAC activation and Stu1 sequestering is triggered concertedly. The sequestering of Stu1 and Slk19 at uaKTs depends on each other. This led to the model shown in Fig. 1n. uaKT localization of Stu1 but not of Slk19 depends on Mps1. We thus suggest that the uaKT localization of Stu1 starts the oligomerization process. CL is the only domain of Stu1 that solely drives sequestering[8]. Furthermore, the interaction with a Stu1/Slk19 oligomer depends on CL. We thus speculate that a conformational change in CL, initiated by the localization of Stu1 to an uaKT promotes the direct or indirect interaction with Slk19 and propagated conformational changes in Slk19 and CL sustain the oligomerization process. In addition, phoshorylation of Stu1 and/or Slk19 by uaKT-localized Mps1 might drive this process, similar to the finding that SAC signaling requires Bub1 phosphorylation in addition to Spc105 phosphorylation[24]. In this case, we would expect that Stu1 and/or Slk19 directly bound to the uaKT (or at least close to it) are the subject of Mps1 phosphorylation, and that this is the final event that starts the oligomerization of Stu1 and Slk19 or at least supports it. In principal, one could also imagine that all Stu1 and/or Slk19 molecules have to be phosphorylated by uaKT-bound Mps1 before they can be subject to sequestering. This would however slow down the sequestering process. Each Stu1/Slk19 molecule would have to contact uaKT-bound Mps1 before it was eligible for sequestering. Otherwise, Stu1/Slk19 could directly bind to the interaction sites that likely increase in numbers during the oligomerization process. As has been described for Slk19[12], also Stu1 facilitates the clustering of uaKTs. Assuming that the oligomers originating from individual uaKTs can form a joined network, the model described (Fig. 1n) provides an attractive explanation for the Slk19 and Stu1-dependent clustering of uaKTs.

Sequestering of Stu1 and Slk19 at uaKTs withdraws these proteins from the spindle and results in the advanced formation of nrMTs. This is primarily the consequence of a compromised mitotic spindle. Only a sequestering defect that resulted in uncompromised spindles (stuΔCL and, with small limitations, Mps1-depleted cells) but not those that exhibited additionally large spindle defects (Δslk19 and Stu1-depleted cells) prevented the advanced formation of nrMTs in comparison with the corresponding WT cells. We therefore propose (Fig. 6b) that the withdrawal of Stu1 and Slk19 from attached KTs and spindle MTs results in shortening of kMTs and collapse of ipMTs, and consequently the freed tubulin allows the formation of nrMTs. As Stu1 stabilizes MTs, its sequestering probably allows nrMTs to be very dynamic and thus ideal to scan the nucleus for uaKTs. Mps1-depleted cells showed an increased formation of nrMTs already in cells without an uaKT. Applying the proposed model, this can be explained by the fact that an Mps1 defect results in monopolar attached KTs[25] and consequently in the shortening of kMTs.

We expect that this mechanism works in metaphase if uaKTs accidentally appear, but particularly in prometaphase when all KTs have to be recaptured (Supplementary Fig. 6). It is unclear what regulates the separation of SPBs from prometaphase to

metaphase and the formation of a metaphase spindle (that competes with the formation of capturing nrMTs). However, when Stu1 is efficiently sequestered at uaKTs this step is clearly prevented. Thus, by sequestering Stu1 and Slk19, uaKTs prevent premature spindle establishment in prometaphase and guarantee the presence of capturing MTs. Thereby, Stu1 sequestering adapts to the severity of the problem and may serve particularly as an emergency strategy for prolonged uaKTs. In agreement with this, we found that cells in prometaphase can exhibit uaKTs with large amounts of sequestered Stu1 and no or little amount of Stu1 at SPBs that then resided in close proximity. As there is no spindle or at best a very small spindle in prometaphase (due to Stu1 sequestering and/or other reasons) it appears feasible that the experimental depletion of Stu1 in early prometaphase will not further enhance the formation of nrMTs, as described[26]. Enhancing the capturing of uaKTs by Stu1 sequestering is, similar to the SAC, not essential for viability. Both appear to be strategies to increase the competitiveness of cells. In this case, these strategies operate in parallel. Thus, if the enhancement of capturing fails (as in stu1ΔCL cells), a KT will remain longer unattached but will not necessarily be lost, as the SAC prevents progression into anaphase. These cells may stay arrested indefinitely or the uaKT is captured after a time delay. This could be the reason why stu1ΔCL cells show only very moderately increased chromosome loss[8]. Another reason could be that the artificial chromosome used in the loss assay is considerably smaller than a WT chromosome. It is unclear how this chromosome behaves during capturing. One might speculate it diffuses easier and thus contacts short MTs at the spindle pole more frequently thus alleviating the need for the long capturing MTs.

In contrast to the results utilizing the KT reactivation assay (Fig. 6a), Stu1 depletion compromised the capturing efficiency if cells were released from nocodazole[6]. This discrepancy may be explained in two ways. (1) Nocodazole treatment frequently produces several uaKTs that form clusters most likely to be due to Slk19/Stu1 oligomerization. As a KT cluster is probably more efficiently captured than a single KT[12], sequestering of Stu1 may also contribute to capturing via cluster formation. (2) KT-delivered Stu1 has a stabilizing role for the capturing MT (Fig. 8). In Stu1-depleted cells, the presence of nrMTs is considerably more pronounced than in WT cells with an uaKT. This may compensate the lacking of MT stabilization in the KT reactivation assay. In nocodazole-treated WT cells, nearly all Stu1 is sequestered (simulating Stu1 depletion) and the spindle is similarly compromised as in Stu1-depleted cells. Upon release, both cells thus experience comparable conditions in respect to timing and abundance of nrMT formation. Lacking MT stabilization in Stu1-depleted cells may thus contribute to the relative capturing defect in Stu1-depleted cells.

Upon lateral interaction of the KT distal to the SPB with the capturing MT, the KT-bound Stu1-Slk19 oligomer partly dissociates from the KT in a majority of the cases and is loaded onto the capturing MT. It is unclear what triggers this effect. In higher eukaryotes, Spc105/KNL1 phosphorylation is disabled when Mps1 dissociates from the KTs upon MT end-on attachment. Lateral KT–MT interaction however is not sufficient for Mps1

**Fig. 7** The fate of uaKT-localized Stu1 and Slk19 after capturing of the uaKT. **a–c** Genotypes of the used strains are listed in Supplementary Table 1. uaKTs were produced in metaphase arrested cells as in Fig. 2 and cells were observed at the indicated timepoints after KT reactivation. Blue arrowheads indicate an uaKT. Bar, 2 μm. **a**, **c** Statistics, see Supplementary Table 3. **a** After capturing of an uaKT Stu1 and Slk19 at least partially dissociate from KTs and colocalize as clusters on the capturing MTs. Only capturing events with at least two Stu1 signals on the capturing MT were quantified. **b** Stu1 clusters deposited after capturing are transported to the SPB along the capturing MT. **c** During the lateral transport of a captured KT a cluster of Stu1 predominantly localizes between the KT and the MT plus end and frequently overtakes the KT shortly before the KT reaches the SPB. Events were analyzed only if the capturing occurred ≥ 1 μm distal to the SPB. Error bars represent the SD of two independent experiments

dissociation[21]. As lateral MT–KT interaction should not suffice to satisfy the SAC, Spc105 probably remains phosphorylated also in *Saccharomyces cerevisiae* under this condition. This may explain why part of the Stu1-Slk19 complex remains at laterally attached

uaKTs. How then is another part of the complex loaded on the capturing MT? The interaction of Stu1 with the MT matrix in combination with KT transport might disrupt the sequestered complex. Stu1 and Slk19 deposited to the capturing MT was

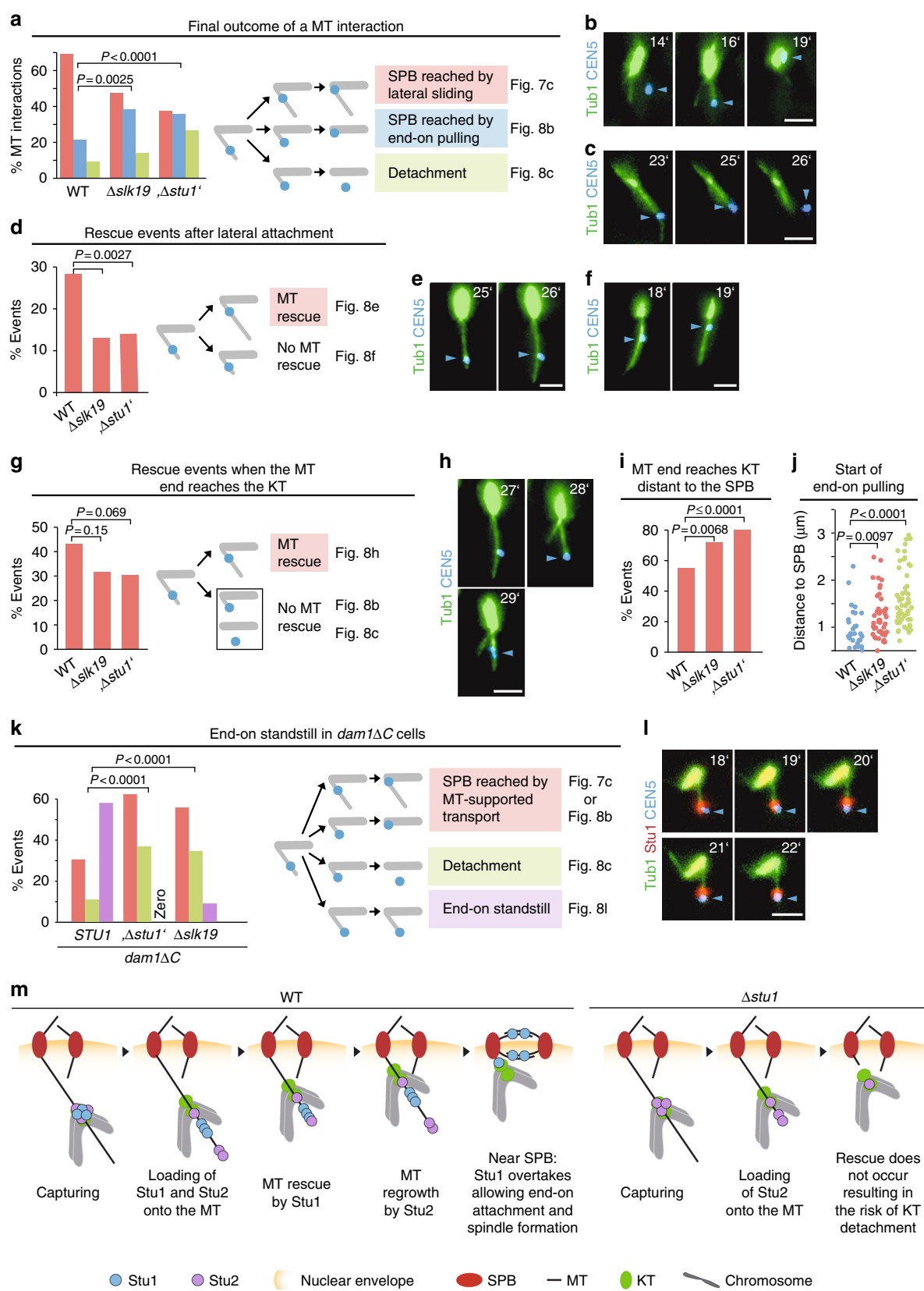

transported to the SPB. The minus-end-directed kinesin Kar3, which probably transports KTs to the SPB[3], appears a likely candidate for this. However, Cin8 and Kip1 have been described to exhibit minus-end directionality under certain conditions[27,28]. A substantial amount of Stu1 resided between the KT and the MT plus end during Stu1 and KT transport. In combination with Stu2[5], this can promote the rescue of a depolymerizing MT before or when the plus end reaches the KT and thus avoid precocious plus-end attachment distal to the SPB (Fig. 8m). The increased numbers of plus-end transports and detachments in Δslk19 or Stu1-depleted cells support this view. Furthermore, the transport of Stu1 appears suspended if the end-on attachment is compromised as in dam1ΔC cells and thus allows Stu1 to prevent MT depolymerization. Frequently the majority of Stu1 overtook the KT when both reached the vicinity of the SPB. This serves well to get the captured chromosome bipolar attached. First, it promotes the formation of the spindle and secondly it allows the depolymerization of the capturing MT and thus plus-end attachment of the KT. Any Stu1–Slk19 complex that was possibly still associated during the lateral transport, dissociated from the uaKTs when they reached the SPB. This is probably due to bipolar or syntelic attachments that reverse the Mps1-dependent Spc105 phosphorylation.

In conclusion, sequestering of Stu1 at uaKTs supports capturing in at least two ways. It evokes the enhanced formation of capturing MTs and stabilizes a MT during the capturing process. What is the relative importance of these two effects? stu1ΔCL cells harboring additional WT STU1 sequester WT Stu1 at uaKTs and relocate it to the capturing MT but fail to compromise the spindle and enhance nrMT formation. The capturing efficiency of these cells was as compromised as that of stu1ΔCL cells that cannot deliver Stu1 to the capturing MT (Fig. 6a). Thus, the enhanced formation of capturing MTs clearly is a prerequisite for efficient capturing and stabilization of a capturing MT will only contribute to the effect if the capturing MTs are present. It is unclear whether similar strategies promote the capturing of KTs in higher eukaryotes. However, it is apparent that there is a global change in the MT network and MT dynamics that facilitates KT capture and spindle formation at the transition from G2 into mitosis[29]. Regulating the localization of MT binding proteins could contribute to these changes. In this respect, it is interesting that the human Stu1 ortholog CLASP1 localizes to KTs most prominently in prometaphase[30].

## Methods

**Strain and plasmid construction.** Yeast strains and plasmids are described in Supplementary Tables 1 and 2. All yeast strains are derivatives of YPH499[31]. Tagging of endogenous genes, promoter replacement and gene disruption was performed by PCR-mediated integration[32].

The following plasmids were integrated into the LYS2 locus to construct the following stu1 and dam1 mutant strains: Plasmid pVS1499 for YJO2818, YJO2723, and YJO2727; plasmids pJO1599, pJO1598, and pCF1563 for YJO2717, YJO2716, and YME2731, respectively; plasmid pME1597 for YME2710 and YME2761; and plasmid pME1561 for YME2726 and YME2782. To produce strains YVS1661 and

YVS2151, a construct containing stu1Δ(aa995-11180)-EGFP-klTRP1 or stu1Δ (aa995-1180)-CFP-kanMX6 was integrated into the STU1 locus as described[8]. To produce YJO2740 (spc105-6A), plasmid pME1546 was constructed by overlapping PCR and integrated into the LEU2 locus.

The auxin-based degron system[33] was used to deplete Stu1, Spc105, Mps1, and Dam1. The corresponding genes were tagged with IAA17 using plasmids pME1595 or pMK43[33], and OsTIR1 was integrated into the ADE2 locus using plasmid pVS1453.

Strains that allowed the rapamycin-controlled Mps1 localization to Spc105 were produced using the plasmids described for the anchor-away method[34].

To produce a strain that allowed the inactivation and reactivation of the KT on chromosome V, a construct carrying the GAL1/10 promoter adjacent to CDEI of CEN3 (in addition to LEU2) flanked by sequences left and right of CEN5 was released from plasmid pMK1459 and used to replace CEN5 by homologous recombination. Tetracycline operator sequences were integrated in the vicinity of the centromere sequence of chromosome V as described[35]. To produce strains that expressed TetR-5xCerulean, the Cerulean sequence carrying an upstream BamHI site and a downstream BglII site was amplified by PCR and multimerized by ligase treatment. Head to tail arrangements were selected for by BamHI and BglII digestion and a fragment containing 5xCerulean was cloned in frame with a pURA3-TetR construct into a pYM25 backbone[32] carrying the hphNT1 marker and the ADE1 terminator. The resulting plasmid (pME1590) was integrated into the terminator sequence of ADE1.

**Fluorescence microscopy.** Live cell imaging was performed on a life science imaging station (CellR; Olympus) with a × 100 Plan Apochromat, numerical aperture 1.4, objective and a charge-coupled device camera (ORCA-ER; Hamamatsu Photonics) at room temperature as previously described[36]. For image acquisition, the Xcellence rt software (Olympus) was used. Images had 15.625 pixels µm$^{-1}$ and a bit depth of 16 bit. Both features were not altered by image processing. Cyan fluorescent protein, Cerulean, green fluoresent protein, and mCherry were used as fluorochromes. For live-cell imaging, cells were resuspended in nonfluorescent medium (NFM, 0.9 g l$^{-1}$ KH$_2$PO$_4$, 0.23 g l$^{-1}$ K$_2$HPO$_4$, 0.5 g l$^{-1}$ MgSO$_4$, 3.5 g l$^{-1}$ (NH4)$_2$SO$_4$, 0.79 g l$^{-1}$ Complete Supplement Mixture + all (MP Biomedicals), 0.5 mg l$^{-1}$ β-alanine, 0.2 mg l$^{-1}$ thiamine HCl, 3 mg l$^{-1}$ Ca-pantothenate, 2 mg l$^{-1}$ inositol, and 0.4 mg l$^{-1}$ biotin). For single images, cells were spread onto microscope slides and seven Z-stacks (0.43 µm) were acquired. For time-lapse imaging used to measure MT dynamics, five Z-stacks (0.35 µm) were acquired and the film tool of the Xcellence software was used. For all other time-lapse experiments, cells were spread onto agarose pads[37] and images were taken at the indicated timepoints (typically starting 7 min after the NFM plus glucose addition). Five Z-stacks were acquired. Typically, three frames were acquired at each timepoint using a motorized stage and the corresponding feature of the Xcellence software.

**Image processing.** All images were projected with maximum intensity using the Fiji software tool "Z-project". Color channels were adjusted individually using the Fiji tool "brightness/contrast". The lookup table was linear.

**Image analysis.** The fluorescence intensity, MT length, and distances between two points were determined using the Fiji tool "Measure".

Intensity of Stu1 fluorescence at the spindle and uaKTs (Fig. 2): The mean intensity of the area of interest was measured, the mean background intensity close to this area was subtracted and the result was multiplied by the area of interest. To correct for photobleaching, the Stu1 fluorescence detected at the spindle of > 20 cells that failed to produce an uaKT was determined for each timepoint and the calculated bleaching factors were used to correct the data obtained for cells with an uaKT. All cells that showed any Stu1 localization at the uaKT (indicating successful KT reactivation) were included in the analysis. This represented 80% of the cells with an uaKT.

Length and number of nrMTs (Figs. 3, 4, 5): The MT length was measured for projected images. Furthermore, the MT length was always determined as the distance between the MT tip and the SPB from which the MT apparently originated although MTs that emit from the spindle region at an angle of about

**Fig. 8** The role of uaKT-localized Stu1 after capturing of the uaKT. **a–l** Genotypes of the used strains are listed in Supplementary Table 1. The uaKTs were produced in metaphase-arrested cells as in Fig. 2 and capturing events were observed at the indicated timepoints after KT reactivation. Statistics, see Supplementary Table 3. **b**, **c**, **e**, **f**, **h**, **l** Micrographs of phenotypes as quantified in **a**, **d**, **g**, **k**. Blue arrowheads indicate uaKTs. Bar, 2 µm. **a–j** By rescuing capturing MTs from depolymerization, uaKT-derived Stu1 diminishes the frequency of end-on attachment and detachment distal to the SPB. **a** Depletion of Stu1 ('Δstu1') and interfering with Stu1 sequestering (Δslk19) increases the frequency of end-on pulling and detachment. **d** The number of rescue events distal to the captured KT is decreased in Stu1-depleted ('Δstu1') or Δslk19 cells. **g** The number of rescue events occurring when the captured KT has encountered a MT plus end is decreased in Stu1-depleted ('Δstu1') or Δslk19 cells. **i** The frequency at which a captured KT encounters a MT plus end distant to the SPB is increased in Stu1-depleted ('Δstu1') or Δslk19 cells. Captured KTs that colocalized with a MT plus end ≥ 0.5 µm distant from the SPB (followed by end-on pulling, MT rescue or detachment) were counted. **j** End-on pulling starts at average at a greater distance from the SPB in Stu1-depleted ('Δstu1') or Δslk19 cells. **k** Stu1 delivered via uaKTs prevents MTs from depolymerizing (standstill) if the plus-end attachment is compromised. **m** Model depicting the role of Stu1 after a MT has captured an uaKT. The role of Stu2 has been included as described[5].

180$^{\text{O}}$ are likely to originate from the more distal SPB. For both reasons the actual values should be higher than the measured ones. Irrespective of this, the measured values serve well to compare the occurrence of nrMTs in different strains. For all strains, the total nrMT length of a cell at a given timepoint was calculated as the sum of all nrMT lengths observed. For WT cells with an uaKT (Fig. 3), the analysis started when the uaKT had acquired 55–70% of the (spindle and uaKT) localized Stu1 and continued till the uaKT contacted a capturing MT or till the end of the film. Different cells were thus analyzed over a variety of time (from 4 to 20 timepoints). In order to exclude the MT stabilization effect observed after KT capture, nrMTs that had clearly engaged in KT capture were omitted from the analysis. WT cells of the identical experiment that failed to produce an uaKT (Fig. 3) were analyzed over 15 consecutive timepoints starting at the beginning of the film. For stu1ΔCL or Mps1-depleted cells (Fig. 4), all cells with an uaKT were analyzed over 4–15 timepoints. Mps1-depleted cells without an uaKT were analyzed over 15 timepoints. Δslk19, Stu1-depleted, and WT control cells (Fig. 5b–d) were analyzed 3.5 h after the shift to the methionine containing medium initiating metaphase arrest.

MT dynamics (Fig. 5e): For cells with an uaKT, all were included if the uaKT had recruited > 50% of the localized Stu1 and had produced nrMTs. The analysis continued until the uaKT contacted a capturing MT or till the end of the film. For Stu1-depleted cells, all were included that produced nrMTs. Growth and shrinkage rates were determined as the MT length difference measured for two consecutive timepoints divided by the time difference and the results were averaged for all cells and timepoints analyzed. Rescue and catastrophe events were counted when the time between start and end of MT shrinkage or growth could be determined. The number of rescue events divided by the sum of time over which all depolymerizing MTs were observed in all cells represents the rescue frequency. The number of catastrophe events divided by the sum of time over which all polymerizing MTs were observed in all cells represents the catastrophe frequency.

Capture efficiency (Fig. 6): The number of cells with an uaKT was determined for the indicated timepoints. A KT was considered as captured if it resided in the spindle region for more than three timepoints.

Capturing (defect) phenotypes (Fig. 8): "SPB reached via lateral sliding" (Fig. 8a) was scored if no end-on attachment occurred during the transport. "SPB reached via end-on pulling" (Fig. 8a) was scored if the KT colocalized with the MT plus end at least one timepoint directly before it reached the spindle pole. "Detachment" (Fig. 8a, k) was scored if the KT colocalized with a MT for at least two timepoints and subsequently lost contact. In about 50% of the counted events lateral or end-on transport previous to the detachment event additionally supported that the KT was indeed attached to the MT. The remaining events were just deduced from MT and KT colocalization. As the analyzed micrographs represent projections of Z-stacks, it appears possible that these scored events include some false positives. However, these errors should occur with similar frequency for the strains analyzed and we thus consider the "detachment" count as valid comparable data. For laterally attached KTs (Fig. 8d), "MT rescue" was scored if a MT grew after the interaction with a KT or did not shrink (for 1 min) and "No MT rescue" was scored if the MT depolymerized after interaction with a KT. For KTs that colocalized with MT plus ends (Fig. 8g), "MT rescue" was scored if a MT grew after a KT reached the MT plus end or did not shrink (for 1 min) and "no MT rescue was scored" if end-on pulling or detachment followed the plus end localization. "SPB reached by MT-supported transport" (Fig. 8k) was scored if lateral sliding or plus-end transport occurred. "End-on standstill" (Fig. 8k) was scored if the KT colocalized with the MT plus end for at least 5 min while no MT shrinkage could be observed or while the MT grew and the KT remained at the plus end.

**Cell culturing**. Cells were grown at 25 °C and 180 r.p.m. Figure 1 (b–l) and Supplementary Fig. 1–2: Cells grown logarithmically in YP medium (yeast extract, peptone) with 2% glucose (Figs. 1b–k) or 2% raffinose, 1% galactose (Fig. 1l) were synchronized in G1 by incubation with 200 ng ml$^{-1}$ α-factor for 2 h. For G1 release, cells were washed with water, suspended in fresh YP medium containing 2% glucose and 15 μg ml$^{-1}$ nocodazole, and analyzed 3 h after the G1 release. For depletion of IAA17-tagged proteins[33], cells expressing OsTIR1 were released from G1 into medium containing additionally 1 mM indole acetic acid (IAA). For Mps1 inactivation in mps1-as1[18] mutants, 1NM-PP1 was added to 10 μM after the G1 release. For Cdc20 depletion (Fig. 1l), cells were shifted to YP medium containing 2% glucose with the G1 release.

Figure 1m: Cells grown logarithmically in YP medium with 2% raffinose and 1% galactose were shifted to YP medium with 2% glucose to deplete Cdc20 and arrest the cells in metaphase. Three hours, later rapamycin was added to 1 μg ml$^{-1}$. One hour later, cells were analyzed in NFM medium containing 2% glucose and 1 μg ml$^{-1}$ rapamycin.

Figures 2–4, 5e, and 6–8: Cells grown logarithmically in synthetic complete (SC) medium (yeast nitrogen base, synthetic complete dropout mixture) lacking methionine and supplemented with 2% raffinose, were shifted to YP medium supplemented with 2% raffinose and 2 mM methionine to arrest cells in metaphase. One hour later, galactose was added to a final concentration of 2% to inactivate the KT of chromosome V. Three to 4 h later, cells were washed two times with water and resuspended in NFM medium containing 2% glucose and 2 mM methionine to reactivate the KT. For Stu1 or Mps1 depletion, IAA was added to 1 mM 3 h after the shift to methionine and cells were analyzed 1.5 h later in NFM medium containing 2% glucose, 2 mM methionine, and 1 mM IAA.

Figure 5a–d: Cells grown logarithmically in SC medium containing 2% glucose and no methionine were shifted to YP medium containing 2% glucose and 2 mM methionine to arrest cells in metaphase, and analyzed 3.5 h later in NFM medium containing 2% glucose and 2 mM methionine. For Stu1 depletion, IAA was added to 1 mM 2 h after the shift to YP medium. Cells were analyzed in NFM medium containing 2% glucose, 2 mM methionine, and 1 mM IAA.

Supplementary figure 3: Cells grown logarithmically in YP medium with 2% raffinose and 1% galactose were arrested in G1 for 2 h, then released into YP medium with 2% glucose (for Cdc20 depletion) and 15 μg ml$^{-1}$ nocodazole and analyzed 3 h after the G1 release. To deplete Spc105, Bub3, and background Stu1, IAA was added to 1 mM 1.5 h after the G1 release. Cells were analyzed in NFM medium containing 2% glucose and if necessary 1 mM IAA.

**Immunoprecipitation and western blot analysis**. Cells were grown logarithmically in YP medium with 2% glucose. When indicated they were arrested with nocodazole for 3.5 h. To deplete Spc105, IAA was added to cycling cells to 1 mM 2 h before the nocodazole arrest. Then proteins were purified as described[8], except that anti-FLAG agarose (Sigma-Aldrich) was used. For western blot analysis, mouse anti-myc (1 : 1,500 dilution; from Covance) and mouse anti-FLAG-horseradish peroxidase (HRP) (1 : 2,000 dilution; from Sigma-Aldrich) were used as primary antibodies and goat anti-mouse-HRP (1 : 10,000 dilution; from Sigma-Aldrich) was used as secondary antibody for the anti-myc antibody.

**Statistical analyses**. The Mann–Whitney Test (Richard Lowry, VassarStats: Website for Statistical Computation, http://vassarstats.net/utest.html, accessed 2016) was used to analyze whether two sets of experimental data were significantly different. It was presumed that the data meet the assumptions of the test (independent and ordinal data). A two-tailed Fisher's exact test (Simon Joosse, In-silico:: Project support for life sciences, http://in-silico.net/tools/statistics/fisher_exact_test, accessed 2016) or Jeremy Stangroom, Social Science Statistics Home Page, http://www.socscistatistics.com/tests/fisher/default2.aspx, accessed 2016) was used to analyze whether defined phenotypes were observed with significantly different frequency in disparate conditions and/or strains. It was presumed that the data meet the assumptions of the test (independent data). If the data set and/or the number of analyzed phenotypes were large the $\chi^2$-test (Jeremy Stangroom, Social Science Statistics Home Page, http://www.socscistatistics.com/tests/chisquare2/Default2.aspx, accessed 2016) was used instead of the Fisher's exact test. It was presumed that the data meet the assumptions of the test (large sample size, independent data). Large sample sizes were chosen to obtain accurate results. To ensure reproducibility, each experiment was performed at least two times. For further information, see Supplementary Table 3.

The representative images in Fig. 3a and Fig. 7b could be observed in three independent experiments. All other images represent the phenotype quantified in the corresponding Figure.

**Data availability**. All relevant data produced by the current work is in this published article including the Supplementary Information or are available from the authors on reasonable request.

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

## Acknowledgements

This work was supported by a grant from the Deutsche Forschungsgemeinschaft.

## Author contributions

C.K., J.O., and J.L. designed experiments and interpreted results. C.K., J.O., M.P., and J.L. analyzed the data and wrote the manuscript. C.K., J.O., M.P., S.N., and V.S. performed the experiments.

## Additional information

**Competing interests:** The authors declare no competing financial interests.

