## [Peer Review File · Nature Communications]

Reviewers' comments:

Reviewer #1 (Remarks to the Author):

In this manuscript, authors expanded their previous findings that Stu1, a CLASP family member, is sequestered at unattached kinetochores. They found that the Stu1 sequestration is carried out by oligomerization with Slk19 triggered by phosphorylation of Spc105 by Mps1. Utilizing the kinetochore reactivation system, the authors beautifully revealed that the Stu1 sequestration at unattached kinetochores enhances the formation of nuclear microtubules that facilitates kinetochore capture. Furthermore, they found that after the capture, Stu1 redistributes from the kinetochores to the capturing microtubules that helps to avoid kinetochore detachment by preventing microtubule shrinkage. The phenotype of the Stu1 sequestration is conspicuous and demonstrable, and their claim that unattached kinetochores promote both the activation of the spindle assembly checkpoint and their own capturing through Mps1 is intriguing, giving insights into the mechanism of efficient kinetochore capture. Therefore, the manuscript is of interest to general readers, even if the molecular mechanism reported here may not be conserved in other organisms. I recommend the authors to address the following points to strengthen their conclusions.

Major points

1. The model that phosphorylation of Spc105 by Mps1 recruits Stu1 for kinetochore capturing in parallel with recruitment of Bub1/Bub3 for the spindle assembly checkpoint is very interesting, although it was addressed only in Figure 1 in nocodazole-treated cells. The model predicts that Mps1 facilitates kinetochore capture, which can be easily examined in the kinetochore reactivation system. Suppression of Mps1 would prevent Stu1 sequestration at the unattached kinetochore while leaving the spindle intact, a situation resembling when *stu1-deltaCL* is expressed. The Mps1 requirement for duplication of spindle pole bodies can be bypassed if Mps1 is depleted after metaphase arrest. As the role of Mps1 in kinetochore capture has not been acknowledged in previous studies (cf. Maure et al, 2007), this point should be clarified in the kinetochore reactivation system.
2. Concerning the model that phosphorylation of Spc105 by Mps1 recruits Stu1, interaction between Stu1 with Spc105 in nocodazole-treated cells should be examined.
3. The relationship between Stu1 and Slk19 is a great breakthrough by the authors and should be stressed more. First of all, direct interaction of Stu1 with Slk19 through the CL domain should be shown by in-vitro binding assay. Another point is that the localization of Stu1 and Slk19 on the spindle and also on kinetochores seems to occur independently, but their oligomerization only occurs in the presence of unattached kinetochores. This suggests a possibility that the binding between Stu1 and Slk19 is regulated through phosphorylation of either proteins by Mps1. The authors need to address the point, before attributing the oligomerization to a conformational change of Stu1 at unattached kinetochores.
4. During lateral attachment, the spindle assembly checkpoint is not satisfied and Mps1 remains on kinetochores. Then why do the Slk19-Stu1 complex disassemble from kinetochores upon lateral microtubule interaction if the complex formation is governed by Mps1? The authors need to explain this point.

Minor points

1. The sequestration of Slk19 to unattached kinetochores seems equally important for efficient kinetochore capturing as its absence on the spindle promotes the formation of random nuclear microtubules independently of Stu1. Therefore, it would be better to stress that this system facilitates the formation of random nuclear microtubules in two ways.
2. Scattering of unattached kinetochores in Stu1-depleted cells shown in Fig. 1c imply that polymerization of Stu1 and Slk19 schematized in Fig. 1n is the mechanism for kinetochore clustering. It is better for the authors to stress more about this intriguing possibility.
3. In Fig. 8, I could not find what do micrographs in e and h mean.

Reviewer #2 (Remarks to the Author):

The current manuscript closely builds on previous work by the same group (G&D 2009, JCB 2014) and investigates the function of the CLASP protein Stu1 for spindle and kinetochore function in *Saccharomyces cerevisiae*. Stu1 displays a very interesting dynamic localization to spindles and kinetochores. It relocates from the spindle to unattached kinetochores, in a manner dependent on the core KT component Ndc80c. At the unattached kinetochore, Stu1 is thought to facilitate the recapturing and subsequent bi-orientation on the mitotic spindle.

Major Points:

1. Many of the key observations regarding Stu1 function at unattached kinetochores have been made by the same group before. The, certainly interesting, novel aspects in this manuscript are: dependence on Mps1/Spc105 phosphorylation for Stu1 localization, dependence on Slk19, effect of Stu1 removal from the spindle on the generation of non-spindle nuclear MTs. Individually, these observations are interesting, but I'm not completely convinced that taken together they gel into coherent insights that go substantially beyond what the authors have already shown in their previous publications. The experiments, as presented are performed in high quality. One caveat is that the interpretation and model almost exclusively rely on phenotypes observed in the CEN reactivation assay. A puzzling aspect here is that the Stu1 Δ CL mutant shows a dramatic effect in the CEN reactivation assay: no accumulation on the uaKT and no removal from the spindle (Figure 2). Yet, this allele is viable, and, moreover, basically does not show any significant increase in chromosome loss (Table 1 in JCB 2014 paper, the authors should also in this paper indicate the cellular phenotypes of the characterized alleles). This illustrates how difficult it may be to relate effects observed in this assay, into the function of the protein during a regular cell cycle.

2. In the absence of any kind of biochemical data, the authors need to avoid any kind of speculation about polymers of Stu1 or co-polymers of Stu1 and Slk19. See particularly page 3, line 67 "three-dimensional array.." or line 76 ".... Slk19 endpoints of an array..", also other instances in the manuscript. Also the model in Figure 1n is inappropriate in the absence of actual experiments that would investigate the biochemical relationship between these components.

3. To me the most interesting novel aspect about the manuscript is the dependence of Stu1 localization to uaKTs on Mps1-mediated phosphorylation of Spc105. Given that recruitment is not dependent on Bub1-Bub3, the only known phospho-MELT "readers", makes this even more intriguing. This seems like an observation that should be extended with further experiments, as it might reveal an interesting coordination between checkpoint function and kinetochore capture.

Minor Points:

1. (Figure 1) The motivation to investigate Slk19 in particular was not completely clear to me. Is the observed behavior specific to Slk19? What about other spindle proteins that contribute to integrity, like Ase1, Bim1, Bik1?

2. (Figure 1): A phenotype that seems to change between the different alleles is the number of additional KT clusters in the noc treated cells (for example high in Slk19delta, or also high in Spc105-6A, if the presented pictures are representative). Maybe this is an important difference.

3. Page 4, line 79, "... the Stu1-Slk19 interaction".. - there's no evidence for a physical interaction in this paper. Please also correct in other parts of the manuscript

4. Page 4 line 97 "... localization change of Ndc80c-bound Mps1... ". This authors should explain this better, it's difficult to follow.

5. The authors frequently use the active verbs like "withdraws" (e.g. page 9, line 226) or "removes" to describe the re-localization of Stu1. I'm not sure that's appropriate, as rather locally separated binding sites with different affinities for Stu1 are created.

Reviewer #3 (Remarks to the Author):

The manuscript by Funk and colleagues unveils a new concept in which unattached kinetochores promote their own capturing. They provide compelling evidence that Stu1, a CLASP orthologue in cerevisiae, together with Slk19, account for this process that is regulated by Mps1 phosphorylation of Spc105. Stu1 is recruited to unattached kinetochores and regulates the dynamics of random microtubules participating in capture. Elegant experiments show a causal relationship between having Stu1 at unattached kinetochores and the efficiency of capturing, upon which Stu1 is relocated to the microtubule and prevents premature microtubule depolymerization before end-on capture. Overall, this is a very sound study, with well-controlled experiments and the conclusions are largely supported by the data. It will most likely be of wide interest and attract the attention of the mitosis field. I have no major issues, only very minor ones, that I would like to draw the attention of the authors:

1- Abstract: "activating the spindle assembly checkpoint". According to the definition of the checkpoint concept by Hartwell and Weinert in the 80s, a "checkpoint" is not activated, but is active by default. Instead, a response is activated due to a checkpoint that, in this case, is sensitive to unattached kinetochores. I would recommend re-writing to reflect this notion.

2- It would be interesting to discuss the data presented here in yeast with what is known about CLASP's function in other systems, including humans. Surprisingly little has been done about the study of CLASP in yeast compared to a quite large breadth of knowledge about CLASPs' function in mammals. For example, it would be interesting to discuss the apparent discrepancies between Stu1 and mammalian CLASPs about putative kinetochore-binding domains (the N-terminal TOGL1 is completely dispensable for kinetochore localization of human CLASPs). Additionally, to what extent the authors believe that the proposed mechanism in this paper might be conserved in humans and what is the experimental evidence for it.

3- The authors refer to "copolymerization" (pages 3 and 4 and Discussion). I struggled a bit with this term and it took me sometime to realize that the authors were not referring to microtubule "copolymerization", but to Stu1-Slk9 oligomerization. Please consider revising the terminology.

4- The authors provide evidence that Stu1 sequestering is independent of several SAC proteins, but failed to test what is probably the strongest candidate to link SAC with microtubule attachment – BubR1/Mad3. Was there a reason not to include Mad3 in their analysis? If not, these data should be added to completely exclude (or not) that there is no link between Stu1 localization and the SAC, other than Mps1.

5- A recent study by the Tanaka group (published while this paper was under consideration at Nature Comms) as proposed a similar model based on similar, yet less extensive findings. While the originality of both works is not being disputed by this reviewer, it is evident that some conclusions are not shared by both studies. For example, in figure 5 of the present study, the authors do report significant differences between the number, length

and dynamicity of nrMTs after Stu1 depletion, which contrasts with what was reported by the Tanaka group. Please discuss.

Response to Reviewers

We thank the reviewers for their very objective and constructive criticism. We have dealt with their comments as follows. Red type is used to indicate the reviewers' comments. Changes in the manuscript are highlighted in yellow.

Reviewer 1:

Major points:

1. The model that phosphorylation of Spc105 by Mps1 recruits Stu1 for kinetochore capturing in parallel with recruitment of Bub1/Bub3 for the spindle assembly checkpoint is very interesting, although it was addressed only in Figure 1 in nocodazole-treated cells. The model predicts that Mps1 facilitates kinetochore capture, which can be easily examined in the kinetochore reactivation system. Suppression of Mps1 would prevent Stu1 sequestration at the unattached kinetochore while leaving the spindle intact, a situation resembling when *stu1-deltaCL* is expressed. The Mps1 requirement for duplication of spindle pole bodies can be bypassed if Mps1 is depleted after metaphase arrest. As the role of Mps1 in kinetochore capture has not been acknowledged in previous studies (cf. Maure et al, 2007), this point should be clarified in the kinetochore reactivation system.

We have included this experiment:

Figures: 2d, 4a and b, 6a.

Text: Results: Chapter 5 (line 146), chapter 6 (line 164, 166, 167-169); chapter 8 (line 196-199); Discussion: Paragraph 1 (line 277, 284-287).

The outcome of this experiment nicely supports our former conclusions.

2. Concerning the model that phosphorylation of Spc105 by Mps1 recruits Stu1, interaction between Stu1 with Spc105 in nocodazole-treated cells should be examined.

We have included an experiment that shows that Stu1 co-purifies with Spc105 when purified from nocodazole-arrested cells.

Supplement Figure 3.

Text: Results: Chapter 2 (line 105-106); Methods: line 534-541

3. The relationship between Stu1 and Slk19 is a great breakthrough by the authors and should be stressed more. First of all, direct interaction of Stu1 with Slk19 through the CL domain should be shown by in-vitro binding assay.

We have decided not to perform this experiment because we consider it very unlikely that we can show CL-Slk19 interaction in vitro. This interaction should be induced only in the presence of unattached KTs. Otherwise we would get Stu1/Slk19 co-accumulation anywhere in the cell. Since the putative interaction is triggered by Mps1-dependent phosphorylation of Spc105 that may or may not have to be part of the KT, we will have to reconstitute the whole process in vitro. This is a plan for the future but would go beyond the scope of this manuscript.

Another point is that the localization of Stu1 and Slk19 on the spindle and also on kinetochores seems to occur independently, but their oligomerization only occurs in the presence of unattached kinetochores. This suggests a possibility that the binding between Stu1 and Slk19 is regulated through phosphorylation of either proteins by Mps1. The authors need to address the point, before attributing the oligomerization to a conformational change of Stu1 at unattached kinetochores.

It is true that phosphorylation of Stu1 or Slk19 by Mps1 could drive their sequestering. We consider it however less likely. Since the effect is triggered by unattached KTs, one would expect Mps1 localized to these KTs to perform the phosphorylation events. The KT-localized Mps1 would then have to access all the Stu1 or Slk19 molecules in order to make them competent for sequestering, which seems difficult, particularly once the process has started. Also, we would not need Spc105 phosphorylation to trigger the effect. We have included these considerations now in the manuscript.

Discussion: Paragraph 1 (line 264-269)

4. During lateral attachment, the spindle assembly checkpoint is not satisfied and Mps1 remains on kinetochores. Then why do the Slk19-Stu1 complex disassemble from kinetochores upon lateral microtubule interaction if the complex formation is governed by Mps1? The authors need to explain this point.

We corrected our data in this respect. Formerly we described that Slk19 completely detaches from uaKTs upon lateral attachment (former figure 7a) and thus concluded that the uaKT-bound Stu1-Slk19 complex dissociates. In the meantime, we found that the 3mCherry tag used in these experiments interferes with Slk19 function. In contrast to Slk19 tagged with GFP or CFP, it severely compromises sequestering of Slk19 to uaKTs upon nocodazole treatment or in the reactivation assay. We therefore repeated the capturing

experiment with Slk19-CFP and found that after lateral attachment, Slk19 co-localizes with Stu1 at the uaKT and at capturing MT sites distinct from the uaKT. We cannot distinguish whether Stu1 and Slk19 that co-localizes with the uaKT, represent MT or KT bound proteins. However, the latter would clearly be in agreement with the fact that Mps1-dependent phosphorylation of Spc105 is most likely maintained during lateral attachment (although KT localization of Mps1 is not sufficient for this in *S. cerevisiae*). We changed the manuscript accordingly and discussed how part of the sequestered Stu1-Slk19 complex could dissociate during lateral attachment.

Figure: 7a

Text: Results: Chapter 9 (line 210-212, 215, 216). Discussion: Paragraph 4 (line 323-333, 348-351).

Minor points:

1. The sequestration of Slk19 to unattached kinetochores seems equally important for efficient kinetochore capturing as its absence on the spindle promotes the formation of random nuclear microtubules independently of Stu1. Therefore, it would be better to stress that this system facilitates the formation of random nuclear microtubules in two ways

We agree. We changed the model accordingly and state this fact throughout the text now.

Figure: 6b

Text: Results: line 204; Discussion: line 274, 281

2. Scattering of unattached kinetochores in Stu1-depleted cells shown in Fig. 1c imply that polymerization of Stu1 and Slk19 schematized in Fig. 1n is the mechanism for kinetochore clustering. It is better for the authors to stress more about this intriguing possibility.

We agree. We have quantified uaKT clustering via the uaKT signals observed in *WT* and Stu1-depleted cells and included this data as Supplementary figure 1. It clearly shows that Stu1 sequestering is required for clustering. Furthermore, we point out that Stu1-Slk19 oligomerization, as proposed in the model, is an attractive way to explain clustering of uaKTs.

Supplementary figure 1

Text: Results: line 78-80. Discussion: line 270-273.

3. In Fig. 8, I could not find what do micrographs in e and h mean.

We apologize for the wrong assignment. The micrographs show representative images that were quantified in d and g respectively. This has been corrected.

Figure 8

Reviewer 2:

General Summary:

The reviewer mentions two points as a summary of our manuscript. Surprisingly, he/she presents these points exactly as they were described in our G&D 2009 paper although the interpretation of these two points have been drastically altered by our new work.

The reviewer states that Stu1 is sequestered “to unattached kinetochores, in a manner dependent on the core KT component Ndc80c.” However, the new work strongly suggests that this just reflects that Mps1 depends on Ndc80c for KT localization and that Mps1-dependent phosphorylation of Spc105 triggers Stu1 sequestering.

He/she also states “At the unattached kinetochore, Stu1 is thought to facilitate the recapturing”. However, we now show that Stu1 has no particular role at the unattached KT. It is the removal of Stu1 from the spindle that guarantees capturing of unattached KTs (see also table below).

By omitting the insights provided by our new data of which there are many (see table below), the tone is set for the claim expressed throughout the review that there are few insights that go beyond our former publications.

Major point 1:

Many of the key observations regarding Stu1 function at unattached kinetochores have been made by the same group before.

We completely disagree, thus let’s compare what we showed before and now.

Previous state	Current state
Stu1 is sequestered at unattached KTs when cells were treated with nocodazole. We had no evidence that sequestering at unattached KTs was able to withdraw Stu1 from intact spindles and affect the spindle structure (since microtubules were destabilized by nocodazole)	Stu1 sequestering is sufficiently effective to deplete Stu1 localized at spindle microtubules and thus to alter the microtubule organization.
We assumed that Stu1 sequestering facilitates capturing. However, the mechanism of this effect was completely unclear. In fact, it was questionable whether Stu1 sequestering or the stabilization of capturing MTs by Stu1 per se facilitates capturing.	We revealed the mechanism how Stu1 sequestering facilitates capturing: 1) Primary effect: Destabilizing or preventing the mitotic spindle leads to the enhanced formation of capturing microtubules which greatly enhances the capturing efficiency. This is a completely novel mechanism. It is also surprising, since a CLASP’s contribution to capturing (if any) was expected to be in the stabilization of all capturing MTs. 2) Secondary effect: Sequestered Stu1,

	deposited onto the MT after capturing, prevents precocious MT depolymerization when KT's are laterally attached.
Stu sequestering depends on the TOGL1 and CL domain of Stu1 and on Ndc80 and Spc105 at the KT.	Stu1 sequestering by unattached KT's is triggered by the Mps1-dependent phosphorylation of Spc105. Furthermore, the ectopic localization of Mps1 to Spc105 activates Stu1 sequestering independent of KT detachment. Stu1 sequestering is thus triggered by the same mechanism as the spindle assembly checkpoint.
	Slk19, another spindle-stabilizing protein, is co-sequestered with Stu1 at unattached KT's

Considering the table above, we are puzzled, as to why the reviewer claims that there is not a large number of new key observations in this manuscript or that there are no new insights.

The, certainly interesting, novel aspects in this manuscript are: dependence on Mps1/Spc105 phosphorylation for Stu1 localization, dependence on Slk19, effect of Stu1 removal from the spindle on the generation of non-spindle nuclear MTs. Individually, these observations are interesting, but I'm not completely convinced that taken together they gel into coherent insights that go substantially beyond what the authors have already shown in their previous publications

I am glad that the reviewer acknowledges here that there are novel interesting aspects. But he/she is not completely convinced that they add up to novel insights. I can just point to the novel insights mentioned above. I'm also disappointed that it is not addressed why the findings are not novel and/or experiments are suggested that would allow us to convince the reviewer. The general statements given do not help in this context.

One caveat is that the interpretation and model almost exclusively rely on phenotypes observed in the CEN reactivation assay.

At least four papers in high ranking journals (Tanaka, K. et al., 2005, Nature 434, 987-94; Tanaka, K. et al., 2007, J Cell Biol 178, 269-281; Gandhi, S.R. et al., 2011, Dev Cell 21, 920-933; Kalantzaki, M. et al., 2015, Nat Cell Biol 17, 421-433) have been based exclusively on this assay. I cannot understand why it becomes a caveat when we are using it. Also, the data revealing how Stu1 sequestering is triggered by unattached KT's does not rely on this assay at all (see Fig.1).

A puzzling aspect here is that the Stu1deltaCL mutant shows a dramatic effect in the CEN reactivation assay: no accumulation on the uaKT and no removal from the spindle (Figure 2). Yet, this allele is viable, and, moreover, basically does not show any significant increase in chromosome loss (Table 1 in JCB 2014 paper, the authors should also in this paper indicate the cellular phenotypes of the

characterized alleles). This illustrates how difficult it may be to relate effects observed in this assay, into the function of the protein during a regular cell cycle.

Deletion of the CL domain in Stu1 indeed results in a strong sequestering defect, no enhanced formation of capturing MTs and a capturing defect. This strongly supports that sequestering of Stu1 at unattached KT's facilitates capturing.

We nowhere claim that the described mechanism is essential (quite the contrary, see our discussion). Likewise, the (intensively studied) spindle assembly checkpoint is not essential. $\Delta mad2$ cells are viable and (depending on the genetic background) exhibit also only a 2.5 fold increase in chromosome loss (Warren, C. D. et al., 2002, Mol Biol Cell 13, 3029-3041). Enhancing capturing and arresting the cell cycle in the presence of unattached KT's are strategies (similar to others like the circadian clock) to increase the competitiveness of cells (particularly those living under wildtype conditions). These strategies frequently also work in parallel. Thus, if the enhanced formation of capturing MTs does not occur (as in *stu1 Δ CL*) KT's will remain longer unattached but will not necessarily be lost since the spindle assembly checkpoint will prevent these cells from proceeding into anaphase. These cells will either stay arrested in metaphase indefinitely (and thus will not show up in the chromosome loss assay) or the KT's will be captured after a delay (and thus suffer a growth disadvantage) because they drift close to the pole or because a small number of capturing MTs always form. Furthermore, the chromosome loss assay is performed with an artificial chromosome that is considerably smaller than native chromosomes. The way this artificial chromosome behaves during capturing is unclear. One might speculate, due to its small size it could diffuse easier and thus contact short MTs at the spindle pole more frequently thus alleviating the need for the long capturing MTs.

Taken together, these points may warrant a discussion in the paper, but the fact that *stu1 Δ CL* cells are viable or show very moderate chromosome loss cannot be a reason to discredit the assay and the conclusions drawn from it.

Major point 2:

In the absence of any kind of biochemical data, the authors need to avoid any kind of speculation about polymers of Stu1 or co-polymers of Stu1 and Slk19. See particularly page 3, line 67 "three-dimensional array.." or line 76 ".... Slk19 endpoints of an array..", also other instances in the manuscript. Also the model in Figure 1n is inappropriate in the absence of actual experiments that would investigate the biochemical relationship between these components.

What we present is just a working model. A working model that best explains the current data. We included it in the result section because it facilitated the understanding of another experiment (Stu1 constructs with and without CL). We can shift it to the discussion section, mention alternatives and point out that there is no direct in vitro interaction shown. To show direct Stu1-Slk19 interaction is difficult. Since the putative interaction is triggered by Mps1-dependent phosphorylation of Spc105 that may or may not have to be part of the KT, we will have to reconstitute the whole process in vitro. This is a plan for the future but would go beyond the scope of this manuscript.

Major point 3:

To me the most interesting novel aspect about the manuscript is the dependence of Stu1 localization to uaKTs on Mps1-mediated phosphorylation of Spc105. Given that recruitment is not dependent on Bub1-Bub3, the only known phospho-MELT “readers”, makes this even more intriguing. This seems like an observation that should be extended with further experiments, as it might reveal an interesting coordination between checkpoint function and kinetochore capture.

Indeed, it is very interesting that checkpoint function and kinetochore capture appear to

be coordinated. This finding is completely new.

We have addressed the fact that the KT localization of Stu1 and Bub1-Bub3 are both dependent on Mps1-mediated phosphorylation of Spc105. We can elaborate on this further in the discussion section. Considering however, that the reviewer does not provide any specific suggestions which aspects we should investigate further, it would be a wild goose chase to figure out which experiments we should perform. Furthermore, to do this thoroughly, it would clearly go beyond the scope of this manuscript.

Minor Points:

1. (Figure 1) The motivation to investigate Slk19 in particular was not completely clear to me. Is the observed behavior specific to Slk19? What about other spindle proteins that contribute to integrity, like Ase1, Bim1, Bik1?

We tested these proteins and others. They are not sequestered.

2. (Figure 1): A phenotype that seems to change between the different alleles is the number of additional KT clusters in the noc treated cells (for example high in Slk19delta, or also high in Spc105-6A, if the presented pictures are representative). Maybe this is an important difference.

The sequestering of Stu1 and Slk19 affects the clustering of unattached KT's. If clustering is compromised one will observe several unattached KT's instead of one or two KT cluster as in WT cells. We can include this information.

3. Page 4, line 79, “... the Stu1-Slk19 interaction”.. - there's no evidence for a physical interaction in this paper. Please also correct in other parts of the manuscript

We will correct this.

4. Page 4 line 97 “... localization change of Ndc80c-bound Mps1... “. This authors should explain this better, it's difficult to follow.

Can be done.

5. The authors frequently use the active verbs like “withdraws” (e.g. page 9, line 226) or “removes” to describe the re-localization of Stu1. I’m not sure that’s appropriate, as rather locally separated binding sites with different affinities for Stu1 are created.

We will correct this.

Reviewer 3:

1- Abstract: “activating the spindle assembly checkpoint”. According to the definition of the checkpoint concept by Hartwell and Weinert in the 80s, a “checkpoint” is not activated, but is active by default. Instead, a response is activated due to a checkpoint that, in this case, is sensitive to unattached kinetochores. I would recommend re-writing to reflect this notion.

We have corrected this phrase to “via”.

Abstract: line 12

2- It would be interesting to discuss the data presented here in yeast with what is known about CLASP’s function in other systems, including humans. Surprisingly little has been done about the study of CLASP in yeast compared to a quite large breadth of knowledge about CLASPs’ function in mammals. For example, it would be interesting to discuss the apparent discrepancies between Stu1 and mammalian CLASPs about putative kinetochore-binding domains (the N-terminal TOGL1 is completely dispensable for kinetochore localization of human CLASPs). Additionally, to what extent the authors believe that the proposed mechanism in this paper might be conserved in humans and what is the experimental evidence for it.

We think that engaging in a general comparison of CLASPs from different systems is not really appropriate for this manuscript that has a very specific theme. Also, it would increase an already long Discussion section even further. We have however included a part that addresses the question whether the mechanism described for *S. cerevisiae* in our work, could exist also in higher eukaryotes.

Discussion: line 360-366

3- The authors refer to “copolymerization” (pages 3 and 4 and Discussion). I struggled a bit with this term and it took me sometime to realize that the authors were not referring to microtubule “copolymerization”, but to Stu1-Slk9 oligomerization. Please consider revising the terminology.

We have exchanged “copolymerization” for “Stu1 and/or with Slk19 oligomerization” or “Stu1-Slk19 oligomerization” throughout the manuscript.

Results: line 82, 91, 102.

4- The authors provide evidence that Stu1 sequestering is independent of several SAC proteins, but failed to test what is probably the strongest candidate to link SAC with microtubule attachment – BubR1/Mad3. Was there a reason not to include Mad3 in their analysis? If not, these data should be added to completely exclude (or not) that there is no link between Stu1 localization and the SAC, other than Mps1.

We are not quite sure, why the reviewer thinks that Mad3 is the strongest candidate, since there is no strong evidence that Mad3 localizes to kinetochores in *S. cerevisiae*. But to exclude the possibility that Stu1 sequestering depends on Mad3 we tested it in $\Delta mad3$ cells. We found that Stu1 was sequestered as in *WT* cells. We have included this data.

Supplemental Figure 2b (former Supplemental Figure 1b)

Text: Results: line 125

5- A recent study by the Tanaka group (published while this paper was under consideration at Nature Comms) as proposed a similar model based on similar, yet less extensive findings. While the originality of both works is not being disputed by this reviewer, it is evident that some conclusions are not shared by both studies. For example, in figure 5 of the present study, the authors do report significant differences between the number, length and dynamicity of nrMTs after Stu1 depletion, which contrasts with what was reported by the Tanaka group. Please discuss.

The reason for this apparent discrepancy is, that in the Tanaka paper, microtubules were observed in early prometaphase. In this phase, the spindle pole bodies are in close proximity, that is, there is no stable mitotic spindle (possibly due to the sequestering of Stu1 and Slk19 at uaKTs). The number of dynamic nrMTs is therefor already high in these experiments and depletion of Stu1 will have no further effect on a (barely existing) spindle and nrMT formation / dynamics. We have addressed this point in the Discussion section.

Discussion: line 291-292.

Reviewers' comments:

Reviewer #1 (Remarks to the Author):

The authors made a considerable effort to address the concerns by the reviewers. Now it is clear that Mps1 facilitates KT capture. However, several concerns remain that need to be addressed.

1. Interaction between Stu1 and Spc105 was shown by an immunoprecipitation assay, detecting myc-tagged Stu1 in the presence or absence of Flag-tagged Spc105. Although I could not find whether these proteins were overexpressed or not, better comparison would be between wild-type Spc105 and Spc105-6A, which excludes the possibility of non-specific binding and further confirms the specific binding of Stu1 to the phosphorylated MELT repeats.
2. A three-dimensional array of alternating Stu1 and Slk19 molecules shown in Fig. 1n is a main model claimed by the authors, but it is too speculative under the current findings. As Slk19 was chosen as a Stu1 partner based on a previous report, but not by a comprehensive analysis of Stu1-binding proteins, it is formally possible that other proteins are also involved. Therefore, interaction between Stu1 and Slk19 should be checked, no matter if it is detected or not, for further discussion. Even if the interaction requires unattached KTs, the authors can still address the point in the immunoprecipitation assay suggested in the previous comment; whether Slk19 co-purifies with Spc105 together with Stu1. By performing the assay in the presence or absence of nocodazole, in this case using anti-myc antibody to precipitate Stu1 instead of Spc105, the requirement of unattached KTs would also be addressed.
3. Considering the dynamic nature of Mps1, the authors' claim that phosphorylation of Stu1 and Slk19 by Mps1 is unlikely would not be a general consensus. Sequential phosphorylation of Stu1/Slk19 in addition to Spc105 by Mps1 is also possible, which is known for SAC components. Although it may be beyond the focus of the current study to pursue the detailed mechanism, the authors should discuss these possibilities.
4. The authors admitted that the enhanced formation of nuclear MTs by Stu1 accumulation at KTs does not have further effect in early prometaphase, the period when virtually all the KT capture occurs in a physiological condition, undermining the significance of their findings. In contrast, a recent study by the Tanaka group showed that Stu1 recruits Stu2 on unattached KTs, which facilitates KT capture by the formation of KT-derived MTs. Thus, a question arises whether the term "sequestration" is appropriate to describe the accumulation of Stu1 on unattached KTs. The authors have to reconsider and clarify the use of the term by citing their paper.

We thank the reviewer for her/his input to further improve our manuscript. Red type is used to indicate the reviewers' comments. Changes in the manuscript are highlighted in blue.

In general, we would like to emphasize that we consider the elucidation of a novel mechanism that allows unattached kinetochores to secure their own capturing as the major achievement of our work. In addition, we think it is very interesting that this mechanism is triggered by the same signals as the spindle assembly checkpoint. We provide abundant information on the details of this mechanism, however it appears unrealistic to expect that we can resolve it to completion within this manuscript. Similarly, the concept of the spindle assembly checkpoint was established more than twenty years ago, but it took many years and publications to reveal its mechanistic details.

In detail, we have dealt with the reviewer's requests as follows:

1. Interaction between Stu1 and Spc105 was shown by an immunoprecipitation assay, detecting myc-tagged Stu1 in the presence or absence of Flag-tagged Spc105. Although I could not find whether these proteins were overexpressed or not, better comparison would be between wild-type Spc105 and Spc105-6A, which excludes the possibility of non-specific binding and further confirms the specific binding of Stu1 to the phosphorylated MELT repeats.

1. In the previous revision, we had performed exactly the experiment the reviewer requested and showed that Stu1 co-purified with Spc105 when cells were treated with nocodazole (that is when Stu1 is sequestered at unattached kinetochores). Both proteins were expressed from their native promoter as indicated by the genotype of the strain used (Supplemental Table 1).

We agree that it would be supportive if we could show that Stu1 does not co-purify with Spc105-6A at unattached kinetochores. However, even after nocodazole treatment the majority of kinetochores is still attached and (relying on the microscopy data) Stu1 remains at the attached kinetochores (and MTs) in nocodazole-treated *spc105-6A* cells (Fig. 1I). We have described before (Funk, C. et al., *J Cell Biol* **205**, 555-571 (2014)) that Stu1 also binds to attached kinetochores albeit under different prerequisites. When we investigated Stu1/Spc105 co-IP in the absence of nocodazole (attached kinetochores) we found that Stu1 also co-purified with Spc105. Moreover, the IP of Stu1 with Spc105 was not markedly enhanced in nocodazole-arrested cells versus cycling cells, indicating that we only detected the more directly bound Stu1 and not the oligomerized Stu1 in that IP. Thus, one can expect that after nocodazole treatment, Stu1 would co-purify with Spc105-6A (of attached kinetochores in *spc105-6A* cells) and with Spc105 of unattached kinetochores in similar quantities. This is in agreement with what we found. Thus, the results of

the IP-experiments are consistent with the fact that Stu1 binds to attached and unattached kinetochores but do not allow a conclusion on whether Stu1 binds to uaKTs in *spc105-6A* cells. The microscopy approach (that provides very conclusive data in this respect) is superior in this case. Nevertheless, we have included the IP data in the manuscript.

Results: line 115-129

Discussion: line 278-286

Supplementary Figure 4

2. A three-dimensional array of alternating Stu1 and Slk19 molecules shown in Fig. 1n is a main model claimed by the authors, but it is too speculative under the current findings. As Slk19 was chosen as a Stu1 partner based on a previous report, but not by a comprehensive analysis of Stu1-binding proteins, it is formally possible that other proteins are also involved. Therefore, interaction between Stu1 and Slk19 should be checked, no matter if it is detected or not, for further discussion. Even if the interaction requires unattached KT, the authors can still address the point in the immunoprecipitation assay suggested in the previous comment; whether Slk19 co-purifies with Spc105 together with Stu1. By performing the assay in the presence or absence of nocodazole, in this case using anti-myc antibody to precipitate Stu1 instead of Spc105, the requirement of unattached KT would also be addressed.

2. Although direct interaction cannot be definitively proven, it would be supportive of our sequestering model if we could show that Slk19 co-purifies with Spc105 or Stu1. We have performed this experiment now but were not able to find co-IP of Slk19 with Spc105 or Stu1. This may indicate that Slk19 interacts with these proteins indirectly or that the applied conditions for cell lysis and immunoprecipitation were incompatible with the stability of the tested interactions, in particular within the oligomerized complex. We favor the latter explanation since we most likely have not detected the oligomerized Stu1 in that IP (see above). We have included these facts in the result section of the manuscript and changed the model in figure 1n. It now includes the information that currently there is no evidence for a direct Stu1-Slk19 interaction. Furthermore, in order to demonstrate that we tested not only Slk19, we included data that shows that several other MAPS (Bim1, Bik1, Stu2, Cin8, Kip1, Fin1, Kar3 and Ase1) are not sequestered at unattached kinetochores and thus are likely not part of the oligomer.

Results: 83-87

Discussion: line 293

Figure 1n

Supplementary Figure 1

3. Considering the dynamic nature of Mps1, the authors' claim that phosphorylation of Stu1 and Slk19 by Mps1 is unlikely would not be a general consensus. Sequential phosphorylation of Stu1/Slk19 in addition to Spc105 by Mps1 is also possible, which is known for SAC components. Although it may be beyond the focus of the current study to pursue the detailed mechanism, the authors should discuss these possibilities.

3. We have changed the discussion in this respect. We absolutely agree that also Stu1 / Slk19 that localizes close to the kinetochore (similar to Bub1) could be phosphorylated by Mps1 in addition to Spc105 and that this may be the final trigger for the oligomerization. We also don't exclude that all Stu1 / Slk19 molecules have to be phosphorylated before they can get integrated into the oligomer. However, we suspect that this would impede the sequestering process, if all molecules have to go through the "bottleneck" of phosphorylation by Mps1 localized at uaKTs.

Discussion: line 295-306

4. The authors admitted that the enhanced formation of nuclear MTs by Stu1 accumulation at KT does not have further effect in early prometaphase, the period when virtually all the KT capture occurs in a physiological condition, undermining the significance of their findings. In contrast, a recent study by the Tanaka group showed that Stu1 recruits Stu2 on unattached KT, which facilitates KT capture by the formation of KT-derived MTs. Thus, a question arises whether the term "sequestration" is appropriate to describe the accumulation of Stu1 on unattached KT. The authors have to reconsider and clarify the use of the term by citing their paper.

4. We were surprised to read this comment. It made us realize that we probably did not make this point clear enough. We did not "admit" that there is no enhanced formation of nuclear microtubules upon Stu1 depletion in prometaphase. It was not our data. We rather reconciled our results with the data from the publication the reviewer mentioned (Vasileva, V. et al., *J Cell Biol* **216**, 1609-1622 (2017)) and that we did cite. We had tried to point out before (in the discussion section of the previous manuscript) that the checkpoint mechanism that we describe (Stu1 sequestering at uaKTs guarantees capturing MTs) should have its main function in prometaphase. Here the spindle pole bodies are in close proximity and there is no or at best a very small spindle. Thus, depleting Stu1 experimentally (as described in Vasileva, V. et al., *J Cell Biol* **216**, 1609-1622 (2017)) cannot create more capturing MTs via spindle depolymerization in prometaphase. It is not clear whether (or how) the SPB separation and the concurrent spindle formation that occurs from prometaphase to metaphase is regulated. (Since Stu1 is essential for metaphase spindle formation, it clearly would be a good candidate for this.) However, if spindle formation could occur while uaKTs are still present, it would deprive these uaKTs of capturing MTs (used for spindle assembly). The

sequestering of Stu1 at uaKTs prevents exactly that. To demonstrate that uaKTs occur long enough in prometaphase to sequester Stu1 we included additional data (Supplementary Fig.5) that quantifies this phenotype. In conclusion, the paper mentioned above does not at all undermine our data and the fact that there is no enhanced formation of nuclear microtubules in prometaphase upon Stu1 depletion can be well explained within the parameters of our model. Also, I cannot find a better word than “sequestering” to describe what happens to Stu1 and Slk19 once there is a prevailing uaKT. We have demonstrated this excessively after nocodazole treatment, kinetochore reactivation and now also in prometaphase of cycling cells. It is not at all comparable to what one finds for other MAPS tested, in particular also not Stu2 (see Supplementary Fig. 1). The role of Stu1 is not just to localize Stu2 to uaKTs. The sequestering of Stu1 at uaKTs has its own important function in kinetochore capture as described above.

We apologize if these points were not made clear enough in our former manuscript. To correct this, we have changed the manuscript and included a model depicting the situation in prometaphase (Supplementary Fig. 6) as well as additional data.

Results: line 224-231

Discussion: line 324-338

Supplementary Figure 5 and 6

REVIEWERS' COMMENTS:

Reviewer #1 (Remarks to the Author):

The reviewer agrees that the proposed mechanism by the authors is very interesting. The authors have responded to all the concerns raised by the reviewer, and sufficient information has now been disclosed for readers to consider the underlying mechanisms.